# Frequency of Bovine Respiratory Disease Complex Bacterial and Viral Agents Using Multiplex Real-Time qPCR in Quebec, Canada, from 2019 to 2023

**DOI:** 10.3390/vetsci11120631

**Published:** 2024-12-07

**Authors:** Sébastien Buczinski, André Broes, Christian Savard

**Affiliations:** 1Département des Sciences Cliniques, Faculté de Médecine Vétérinaire, Université de Montréal, 3200 rue Sicotte, Saint-Hyacinthe, QC J2S 2M2, Canada; 2Biovet Inc., Division of Antech Diagnostics and Mars Petcare Science & Diagnostics Company, Saint-Hyacinthe, QC J2S 8W2, Canada; andre.broes@antechdx.com (A.B.); christian.savard@antechdx.com (C.S.)

**Keywords:** bronchopneumonia, infection, bovine pathogens, BRDC, bacteria, viruses

## Abstract

The bovine respiratory disease complex (BRD) is caused by various bacteria and viruses. The objective of this study was to report the proportion of detection for 10 different bovine respiratory infectious agents (6 viruses and 4 bacteria) using two real-time qPCR assays in Quebec, Eastern Canada, during a 5-year period. The majority of the 1875 samples was positive for at least one agent, with bacterial agents more commonly retrieved than viruses. Pathogen detection depends on the type of sample and season. More pathogens were found in samples pooled from different individuals vs. individual samples. More pathogens were found in the winter and fewer in the summer. This study gives interesting perspectives to improve the diagnosis of this condition in cattle.

## 1. Introduction

The bovine respiratory disease complex (BRD) is one of the major infectious diseases in cattle, which is mainly clinically represented by the presence of fever associated with various respiratory signs and commonly associated with bronchopneumonia. Various viral and bacterial pathogens can cause BRD [1]. Some pathogens are generally not found in healthy animals (such as the bovine respiratory syncytial virus, BRSV), whereas others can be found in the respiratory tract of healthy animals (such as *Pasteurellaceae*) but could invade the lung in the presence of any other environmental stressor or host risk factors, as well as viral infection [1,2,3].

Investigation of the etiology of BRD is an important step when facing an outbreak of bronchopneumonia or any other form of respiratory infection to document the presence of specific agents in order to adapt therapeutic protocols and target adequate preventive measures. Various samples can be used for these analyses, such as, among others, nasal or deep nasopharyngeal swabs, bronchoalveolar lavages, and transtracheal aspirates [2]. Nowadays, because of their simplicity and very high agreement with tracheal and bronchial sample results [4], double-guarded deep nasopharyngeal swabs are commonly used by practitioners in calves [5]. Various methods can be implemented for the detection of viral and bacterial components of the clinical outbreak under investigation [2,6]. In the last decade, real-time qPCR (RT-qPCR) detection of viral and bacterial nucleic acids targeting multiple BRD pathogens (multiplex PCR probes) has emerged as a fast way to detect the presence of bacterial and viral pathogens in various samples [7,8,9]. The positive fluorescing signal for the identification of each targeted pathogen can be observed in real time during thermocycling PCR amplification: the lower the cycle threshold when fluorescence is observed, the higher the targeted pathogen nucleic load (copies per mL). Therefore, reporting cycle threshold (Ct) values is an interesting semi-quantitative way to interpret the potential pathogen load in a submitted sample, even if it cannot be extrapolated from one laboratory to another.

The use of various multiplex RT-qPCR kits for bovine infectious disease has therefore grown because of the rapidity of obtaining a test result compared to conventional testing strategies that are based on culture (for bacteria), virus detection, or antibody responses (e.g., serology). The simultaneous testing of various pathogens is also an incentive that makes the test cost per pathogen lower than individual pathogen detection methods. Research on BRD PCR multiplex results of clinical samples obtained from various field settings is scant. Disconnection between laboratories and clinicians has also emerged as an important topic in the medical field [10,11]. The added value of presenting results obtained from various clinical settings is an important incentive to strengthen the link between the clinician and the laboratory. This would ultimately improve the way the submitted samples are labelled with minimal, pertinent clinical information, which would help build new knowledge based on passive surveillance of laboratory-accumulated data [11].

The objective of the current study was, therefore, to report the results from a 5-year period on submitted samples of a 10-pathogen multiplex RT-qPCR used in a diagnostic laboratory (Biovet Inc.) in the Province of Quebec, Canada, as well as the associations between the pathogens found. Our main hypothesis was that the frequency of the different BRD agents would vary and that multiple agents would be commonly found in the same sample. We also wanted to investigate potential differences between the results of pooled and individual samples.

## 2. Materials and Methods

### 2.1. Data Acquisition and Sample Preparation

The data collected from clinical samples submitted to the Biovet laboratory (Biovet Inc., Saint-Hyacinthe, QC, Canada) were obtained from 1 January 2018 to 31 December 2023. The RECORD guidelines for reporting observational studies using routinely collected health data were followed [12]. Unfortunately, because of variability in terms of the sample submission forms completed by the referring veterinarians, it was not possible to know more about the specific context of submission (surveillance vs. clinical outbreak of respiratory problems), the age or breed of the cattle from which the samples had been obtained, and the type of sample submitted. Information was only obtained on whether the sample had been taken from a single individual as opposed to being a pooled sample.

The sampling season was defined similarly to a recent British study [13]. December, January, and February were grouped for the winter period, March, April, and May for the period spring, June, July, and August for the summer period, and September, October, and November for fall.

The samples were prepared as described previously [14]. Briefly, a volume of 500 μL of PBS was added to the swabs and then vortexed. The nucleic acids were then extracted from a 200 μL suspension using a nucleic acid purification kit (MagMAX^TM^, Pathogen RNA/DNA Kit, Thermo Fisher, Toronto, ON, Canada) and the automated KingFisher^TM^ Flex Purification System (Thermo Fisher, Toronto, ON, Canada), following the manufacturer’s recommendations and elution with 90 μL of nuclease-free water. The same method was applied to obtain pooled samples from individual samples at the clinician’s request for the analysis of a pool of such samples.

### 2.2. Analysis and Interpretation of Results

Samples were examined for major bovine viral pathogens, including bovine herpesvirus type 1 (BHV1), bovine respiratory syncytial virus (BRSV), bovine parainfluenza type 3 virus (BPI3V), and bovine coronavirus (BCV), as well as bacterial) respiratory pathogens (*Mannheimia haemolytica*, *Pasteurella multocida*, *Histophilus somni*, and *Mycoplasmopsis bovis*), using Bovichek^®^ MRB bacteria qPCR and Bovichek^®^ MRB virus qPCR kits (Biovet, Saint-Hyacinthe, QC, Canada). Testing was performed according to the manufacturers’ instructions. The samples were examined for influenza type D virus (IDV) using the primers and probe previously described [14]. The samples were also examined for the presence of BVDV using Bovichek^®^ BVDV-BDV PCR (Biovet, Saint-Hyacinthe, QC, Canada), as previously reported [15,16].

The interpretation of the test results was based on the value of Ct required to obtain a positive signal. The value of Ct is negatively associated with the number of genetic copies in an analyzed sample: the higher the Ct, the lower the number of genetic copies in the sample. Therefore, the Ct values were interpreted, as given by veterinary practitioners, as negative (Ct ≥ 35), weak (35 < Ct ≤ 30), moderate (30 < Ct ≤ 25), highly positive (25 < Ct ≤ 20), and strongly positive signals (Ct < 20) for the respiratory pathogens. Due to the fact that several BRD pathogens are also found in commensal respiratory flora of healthy cattle, the interpretation of a positive test could potentially be interpreted differently based on the pathogen detected and its level of Ct. The general equation of correspondence between the number of genetic copies and Ct is as follows: Log(DNA or RNA copy) = (Ct − B)/M, where Ct is the cycle threshold value to reach a positive signal, B is the negative threshold (here 35), and M is the reaction slope for a 100% efficient reaction [17]. However, it is important to mention that there are currently no widely validated benchmarks to interpret Ct objectively in terms of pathogen values because of interlaboratory variability.

### 2.3. Statistical Analyses

All analyses were performed using R statistical software version 4.4.1 [18]. Data wrangling was performed using the “tidyverse” packages suite [19]. It was possible to know whether a sample had been obtained from an individual as opposed to being a pooled sample. Information regarding the number of animals per sample (individuals in the pool) was collected.

The raw Ct results were then used to determine the correlation of excretion among the different pathogens included in the multiplex test using the non-parametric Spearman rho correlation coefficient [20]. The interpretation benchmarks for correlation coefficients were based on a priori reported benchmarks [21]. Briefly, correlation was considered negligible between 0 and 0.10, weak between 0.10 and 0.39, moderate between 0.40 and 0.69, strong between 0.70 and 0.89, and very strong between 0.90 and 1.00.

As a complementary approach to consider Ct as a continuous variable with Spearman’s rho when the frequency of isolation was low, Chi-squared tests (with Yates continuity correction when relevant) were used to determine the relationship between each pair of pathogens using the clinical positivity threshold (Ct < 35 considered as positive vs. negative if Ct ≥ 35). The strength of the association between the variables was calculated using the Phi (φ) coefficient and associated benchmarks [22]. Briefly, φ < 0.10 was considered as a trivial association, φ between 0.10 and 0.29 as a small association, φ between 0.30 and 0.49 as a medium association, φ between 0.5 and 0.69 as a large association, and φ ≥ 0.70 as a very large association. Chi-squared analyses were also performed to look for the proportion of agent-specific positive samples depending on the year, season, and individual or pooled samples. To account for multiple tests and limit the detection rate of false positives, the significance threshold was set, a priori, to *p* < 0.005. This is generally considered a good compromise when multiple independent tests are performed [23]. Beyond reporting the frequency of samples positive for a particular agent, we also reported the proportion of samples with two or more bacteria, the proportion of samples with two or more viruses, and the proportion of samples simultaneously positive for one or more viruses and one or more bacteria. Finally, the proportion of samples positive for two or more viruses (but not positive for bacteria) and the proportion of samples positive for two or more bacteria (but not positive for viruses) were also compiled.

To explore the association between the count of the total number of pathogens found and the year, season, and number of individuals within a sample, a Poisson regression analysis was performed. A full model was offered with potential explanatory variables (i.e., year, season, and number of individuals (one vs. two vs. three or more) in the analyzed sample), which also included potential interactions between pooled sample and season as well as season and year. Manual backward model selection was performed until all remaining explanatory variables had a *p*-value <0.05. Model fit and risk of overdispersion were checked using the “DHARMa” R package [24]. The same models were used on the number of bacteria and viruses, respectively, following the same approach, in order to understand the relative share of each BRD component in the general model.

## 3. Results

A total of 1875 samples were obtained during this 5-year period, with 319, 343, 352, 441, and 420 samples for year 2019, 2020, 2021, 2022, and 2023, respectively. A majority of samples had been submitted in the winter (31.09%, *n* = 583), followed by fall (25.92%, *n* = 486), spring (24.64%, *n* = 462), and summer (18.35%, *n* = 344). The distribution of pooled samples and season submission is indicated in Figure 1. The samples submitted varied over the seasons and years: for example, pooled samples increased over the years (Chi-squared tests *p* < 0.005). Most submitted samples were individual samples, as indicated in Figure 1 (82.51%, *n* = 1547).

A total of 19.31% (*n* = 362) of the samples did not present any pathogen, whereas two or more pathogens were found in 54.1% (*n* = 1014) of the samples, as shown in Figure 2. The frequency of pathogens found by sample is indicated in Table 1. Among the viruses, BCV was the most commonly found (27.5% of samples, *n* = 516), followed by BRSV (17.7%, *n* = 332), whereas, for bacteria, *Pasteurella multocida* (50.1%, *n* = 940) and *Mannheimia haemolytica* (26.9%, *n* = 505) were the most common. The distribution of Ct and the correlation between Ct agents are presented in Figure 3a,b. The three strongest correlations between Ct agents were observed between BHV1 and BPI3V (rho = 0.273), *Histophilus somni* and *Mycoplasmopsis bovis* (rho = 0.223), and *Pasteurella multocida* and *Mycoplasmopsis bovis* (rho = 0.211). When comparing the positivity of the agents (as dichotomous variables), the highest correlation was seen between the same agents, but the strength of correlation was considered small, as indicated in Figure 3c.

The proportion of positive samples for multiple viruses, multiple bacteria, and both bacteria and viruses was 10.0% (*n* = 188), 36.0% (*n* = 675), and 37.2% (*n* = 698), respectively, as shown in Figure 4. Among the samples only presenting multiple viruses (1.5%, *n* = 29), the majority (86.2%, *n* = 25) had only two viruses, while three and four viruses were found in three (10%) and one sample (3.5%), respectively. In samples only having multiple bacteria (15.3%, *n* = 287), a majority had two bacteria (*n* = 195, 67.9%), while 76 (26.5%) and 16 (5.6%) samples had three and four bacteria, respectively.

The various combinations of pathogens in the test result profiles are summarized using an upset plot (Figure 5).

The associations between individual pathogen detection and year, season, and pooled samples are indicated in Table 2. No particular association was seen for the yearly distribution of pathogens. However, an increased proportion of several viral (BRSV, BCV, and IDV) and bacterial pathogens (*Mycoplasmopsis bovis* and *Pasteurella multocida*) was observed during the winter period (Table 2). In pooled samples for which the number of individual animals making up the pool had been indicated (95%, 312/327), 153 samples contained samples from 3 or more specimens (63.3%, *n* = 97 samples). Up to seven individuals (*n* = 3 samples (2.2%)) could be pooled in a sample. Individual samples were associated with a lower proportion of BRSV, BCV, and *Mannheimia* haemolytica compared to the pooled samples.

The number of pathogens isolated per sample and season of sampling was linked to the total number of pathogens isolated using a Poisson multivariable model (Table 3). Increased incidence rates of detection were found for pooled samples and for samples taken in the winter as opposed to fall, whereas the incidence rates were decreased in the summer compared to fall. This was mainly due to the influence of the increased incidence rate of virus detection during the winter period.

## 4. Discussion

Rapid etiological diagnosis is of utmost importance when facing a respiratory outbreak in a herd. The current study reported the agents obtained from commercial multiplex real-time PCR tests using a large number of respiratory samples in the Province of Quebec, Canada. The results indicated that, among the components of BRD viruses, BCV and BRSV were the two most commonly found viruses (found in 27.5% and 17.7% of cases, respectively), whereas *Pasteurella multocida* was by far the most common bacterial pathogen (found in half of the samples), almost twice as prevalent as *Mannheimia haemolytica* (26.9% of the samples), *Histophilus somni* (22.6% of the samples), or *Mycoplasmopsis bovis* (22.4% of the samples). The mixed detection of viruses and bacteria or multiple bacteria was observed in 37.2% and 36.0% of samples, respectively, when using the default positivity threshold of Ct < 35, whereas multiple viruses were detected in 10.0% of samples.

Overall, 80.7% of samples yield at least one positive infectious agent, which is similar to the percentage (82.7%) obtained from nasal swabs of 133 sick cattle in a Slovenian study [8] and the 77.6% found in the majority of deep nasopharyngeal swabs in a UK study [13]. Despite the fact that the presence of mixed infectious agents is a commonly accepted pattern in the pathophysiology of bronchopneumonia [1], we could not find a strong correlation of pathogen-shedding patterns in the analyzed samples. Co-infection could generally occur in a sequential pattern, with a primary viral or bacterial pathogen which is co-infected or superinfected by other agents [1]. This could potentially change the dynamics of respiratory pathogen-shedding depending on the pattern of co-infection and the stage of the disease.

Shedding may also depend on the agents involved, but this could not be investigated further in the current study due to the one-off nature of the samples available. Interestingly, bacterial agents were generally found more frequently than viral agents (with the exception of BRSV and BCV), which is similar to previous studies reporting multiplex PCR results of both types of BRD agents in respiratory samples in Quebec [14], Europe [8,13,25], and Japan [7]. However, it is difficult to compare our study findings due to regional differences and production system particularities, as well as the type of samples submitted.

Among the viral pathogens, BCV was the most commonly reported virus (found in 27.5% of the samples using the default positivity threshold of Ct < 35), followed by BRSV (17.7%) and IVD (4.6%). BCV has emerged in the last 30 years as a component of respiratory tract infection problems in cattle, in addition to its enteric tropism. The difference between enteric and respiratory BCV strains remains controversial, as recently reviewed [26,27]. BCV can also be isolated from the respiratory tract of healthy calves, which adds to the controversy of respiratory BCV pathogenicity as a primary agent [28,29,30]. On the other hand, intranasal challenge of naive calves by BCV results in diarrhea and mild clinical respiratory signs such as nasal discharge, cough, and, less commonly, fever (5 out of 15 challenged calves), mostly associated with upper respiratory tract infection and bronchial/bronchiolar epithelial damage, with no direct pathogenic effect on the lung parenchyma [31]. A sound, consensus interpretation of a BCV-positive result is required, however, to determine the real role of BCV in the respiratory tract, especially when other respiratory pathogens are also present.

This contrasts with BRSV, which is generally considered a primary pathogen since it is not found in healthy animals, except in low shedding quantities in cattle recently vaccinated with modified live vaccine [32]. Therefore, keeping recent vaccination history in mind, positive PCR detection of BRSV is compatible with the implication of the virus when sampling is performed to investigate a respiratory outbreak. We found 17.7% BRSV-positive samples (in the mid-range of other studies’ findings [7,8,13,14,25]), a majority of them associated with a high to very high positive result (Ct < 25 (31.3% positive samples) and Ct < 20 (26.8%)). The frequency of BRSV-positive samples was lower in a previous Quebec study (13.9% of 883 nasal or nasopharyngeal swabs) in calves with respiratory signs between 2017 and August 2020 [14], roughly similar to the work by Enholm et al. [13] in the UK, who reported 21.6% positive samples in clinically affected calves, but lower than Slovenian and Japanese studies, which found 40.6% (54/133) and 32.6% (29 out of 89) positive nasal swabs from clinically affected calves [7,8].

Interestingly, IDV was the third detected virus, found more commonly than BPI3V, BHV1, and BVDV. IDV was discovered relatively recently in 2011 from swine samples [33]. IDV can cause mild respiratory disease during experimental challenges in cattle [34], emphasizing its role as a potential primary or co-factor of BRD. A previous Quebec study showed a similar frequency of 5.3% positive samples in a total of 883 nasal or deep nasopharyngeal swabs [14]. When analyzing the positive samples in our study in greater depth, we found that 50% of positive IDV samples were weakly positive, with Ct between 30 and 35, therefore indicating relatively low shedding levels at the time of sampling. Circulation of IDV is evident in North America [35], and the virus can also be recovered from nasal lavages of dairy workers in large US dairy herds [36]. However, type D influenza virus is not considered zoonotic, as opposed to type A influenza, including, for instance, the highly pathogenic H5N1 influenza virus recently observed in US farms [37]. BPI3V was only found in 4.3% of the samples, whereas BHV1 (2.7%) and BVDV (0.9%) were less commonly found, which is similar to previous reports [8,14].

The most common bacterium found was *Pasteurella multocida*, which was twice as frequent as other bacteria. *Pasteurella multocida* was retrieved in deep nasal swabs of healthy animals, with 9% to 17% of calves < 10 weeks old shedding the bacteria in beef and dairy farms in the UK, respectively [38] (using bacterial cultures of deep nasal swabs). These previous percentages were close to what was reported in <2-week-old healthy calves (based on clinical and lung ultrasound findings) from 20 Quebec dairy herds (13.2% out of a total of 159 sampled calves) but much lower than the values found in older calves (4–8 weeks old), with a prevalence of 43.8% of said calves (out of a total of 64) shedding *Pasteurella multocida*, detected by bacterial cultures of double-guarded deep nasopharyngeal swabs [39]. In the same study, the proportion of *Pasteurella multocida*-positive samples increased to 71.4% in calves with clinical pneumonia (clinical score positive and presence of ultrasonographic lung consolidation). When comparing our study findings to studies with multiplex real-time PCR results from sick animals, our findings were similar to those of previous studies [7,8,14,40] but much lower than the values found in Denholm et al. [13], who reported 93.4% positive samples for this pathogen. The frequency of positive samples varies widely for other BRD bacteria (*Mannheimia haemolytica*, *Mycoplasmopsis bovis,* and *Histophilus somni*) in studies reporting multiplex PCR tests [7,8,13,14,25,40]. The current study’s findings of a relatively low *Mannheimia haemolytica* frequency of positive samples were compatible with estimates from previous studies in Québec [14] and Denmark [25] regarding *Mycoplasmopsis bovis* and *Histophilus somni*, close to the mean range of our reported frequency [7,8,13,14,25,40].

The use of pooled samples is a common clinical practice to decrease the costs of PCR examination [13]. This practice can be considered a practical way to reduce the costs of analyses and increase the chances of finding pathogens that are not shed by all affected animals during a herd outbreak. The challenges associated with pooling samples have been discussed in various contexts, trying to reach the optimal number of patients to pool without altering the sensitivity of pathogen detection [41]. To date, there is no consensus on reliable methods and guidelines for pooling samples and interpreting the associated PCR testing in cattle with BRD [2]. In human medicine, pooling samples has been deemed a reliable method for the detection of SARS-CoV 2 from pooled nasopharyngeal and oropharyngeal swabs transported in 3 mL viral transport medium [41]. A total of 35 pooled samples (*n* = 8 samples) were compared mixing 0 (*n* = 13), 1 (*n* = 11), 2 (*n* = 5), 3 (*n* = 5), and 4 (*n* = 1) positive samples (mean Ct of 30.4 to 34.2 for positive samples). The comparison between the individual results used as the gold standard and the pooled results (positivity threshold of Ct ≤ 40) was very good, with a sensitivity of 95.4% and a specificity of 100%. However, a decrease in the mean Ct of the pooled samples was observed (mean −3.4 Ct decrease, with an extremum of around −9 Ct) [41]. Although the interpretation of the absence vs. presence of a specific pathogen could be useful for abnormal inhabitants of the respiratory tract in the absence of recent modified live vaccination, it is more challenging to interpret for pathogens that could be found in healthy animals. For example, a positive BRSV sample is generally considered a significant finding, whereas *Pasteurella multocida* can be found in healthy animals [42]. We could not directly compare pooled sample results to individual sample results due to the study design. Therefore, our findings should be taken with caution. However, it is interesting to note that the observed proportion of positive samples increased for BRSV, BCV, and *Mannheimia haemolytica*. An increased rate of detection of viral respiratory pathogens was reported in pooled nasal samples (2–6 swabs per pool) compared to individual samples in a previous Irish study for BRSV (17.7% vs. 9.7%), BCV (36.0% vs. 19.9%), BPI3V (13.2% vs. 4.7%), and BHV1 (10.6% vs. 5.3%), but not for BVDV (6.1% vs. 4.1%) [43]. An increased detection rate of some BRD bacteria should not be surprising since a recent Danish study using high-throughput RT-PCR showed that mixing one positive individual nasal sample (Ct < 20) with up to nine negative nasal swabs only resulted in a slight decrease in the pooled Ct (for the BRD pathogens tested (i.e., *H. somni* and *M. bovis*)) [44]. This needs to be investigated in terms of practical applications for lower shedding levels. Recommendations on the expected use and indication of pooling strategies should also be further investigated using a prospective study design, directly comparing individual vs. pooled samples in various clinical contexts. The interpretation of Ct and its correlation with viral or bacterial loads may also depend on the test used and the laboratory, requiring further studies.

The BRD complex is generally associated with various risk factors that are also animal stressors, such as transportation, stressful procedures, and environmental changes. Wide variations in the weather during seasonal changes are important stressors for calves [45]. There were seasonal trends observed in the current study. The multivariable Poisson regression model showed that the number of positive agents per sample was higher in the winter and lower in the summer when compared to results in the fall. Interestingly, this was mostly associated with the effect of the number of viral BRD agents found. The viral BRD agents involved in respiratory problems are more commonly found in cold conditions [43,46,47]. However, it is difficult to determine the relative part played by local changes in weather conditions in increased host stress as opposed to an increased efficacy of viral transmission, as shown for human respiratory syncytial virus, influenza, and parainfluenza viruses, which are known to spread easier at cold temperatures [48].

This study was based on data retrieved from field samples sent to a particular laboratory, and this also comes with particular limitations. The vast majority of samples analyzed were samples from clinical BRD outbreaks, because they came from private practice investigations. However, we could not investigate in depth the specific reason for submitting a sample, the type of bovine production and sample, or the characteristics of the BRD outbreak. A majority of samples were nasal or double-guarded deep nasopharyngeal swabs due to geographical preferences [5]; unfortunately, it was not possible to assess this characteristic further. Moreover, no information was available concerning the herd’s recent history of vaccination with modified live vaccines, a factor which can generally lead to a low level of pathogen shedding [32]. We consider that this is unlikely to have occurred in our current study since veterinarians are generally aware of this information. These limitations come with using data not primarily collected for research purposes [49]. The systematic entry of minimal, specific parameters regarding field samples into laboratory databases (including the age of the animal, the type of sample, the vaccination history, and minimal information on the type of outbreak) could help add value to the different results and associations between the pathogens found.

## 5. Conclusions

This observational study, which gathered numerous respiratory samples of sick animals, showed the concomitant occurrence of multiple pathogens in the same samples. Despite some limitations due to the study’s observational nature, seasonal associations with some pathogen detection rates were observed, whereas an increased proportion of positive results from pooled samples was shown for BRSV, BCV, and *Mannheimia haemolytica*. Interestingly, the total number of pathogens increased in the winter, especially due to increased virus recovery numbers. Pooling the samples was also associated with the detection of more pathogens. These findings should be confirmed in future prospective investigations.

## Figures and Tables

**Figure 1 vetsci-11-00631-f001:**
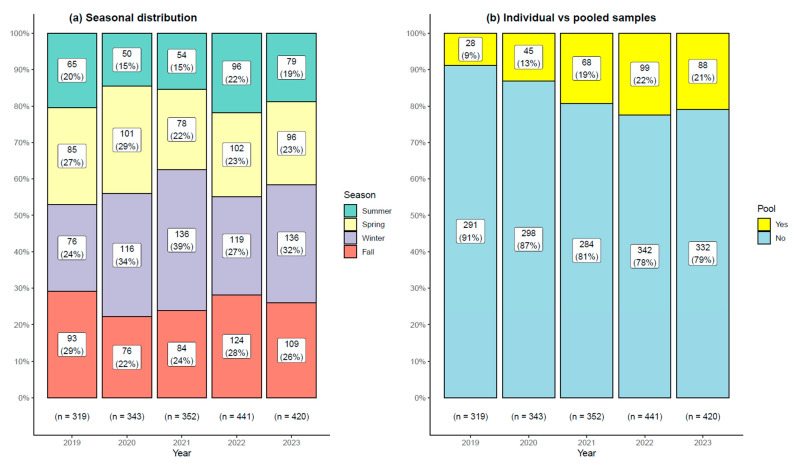
**Comparison of seasonal distribution and type of samples submitted to the Biovet laboratory during a 5-year period** (***n* = 1875**)**.** The seasonal distribution (**a**) was obtained by grouping September, October, and November for fall, December, January, and February for winter, March, April, and May for spring, and June, July, and August for summer. The distribution of pooled vs. individual analyses is also presented in (**b**).

**Figure 2 vetsci-11-00631-f002:**
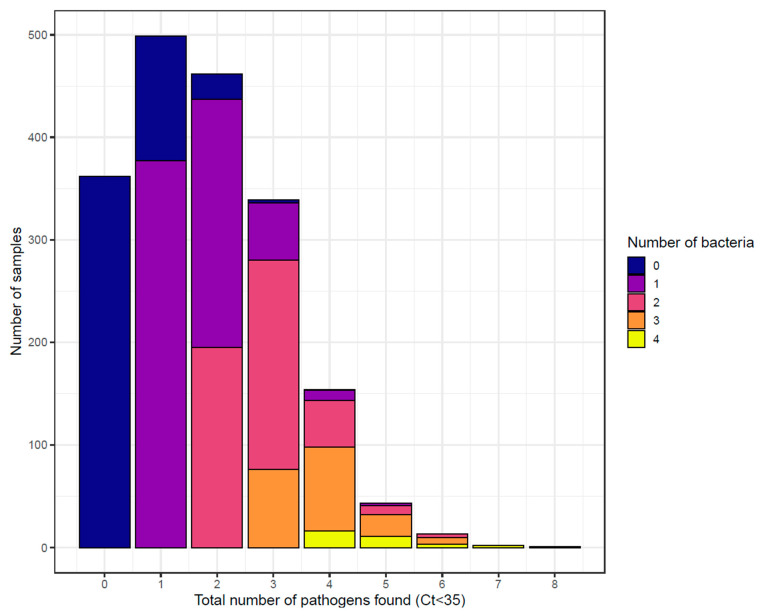
**Distribution of the total number of pathogens found per sample** (**Ct < 35**) **and the number of different bacteria found** (***n* = 1875**)**.**

**Figure 3 vetsci-11-00631-f003:**
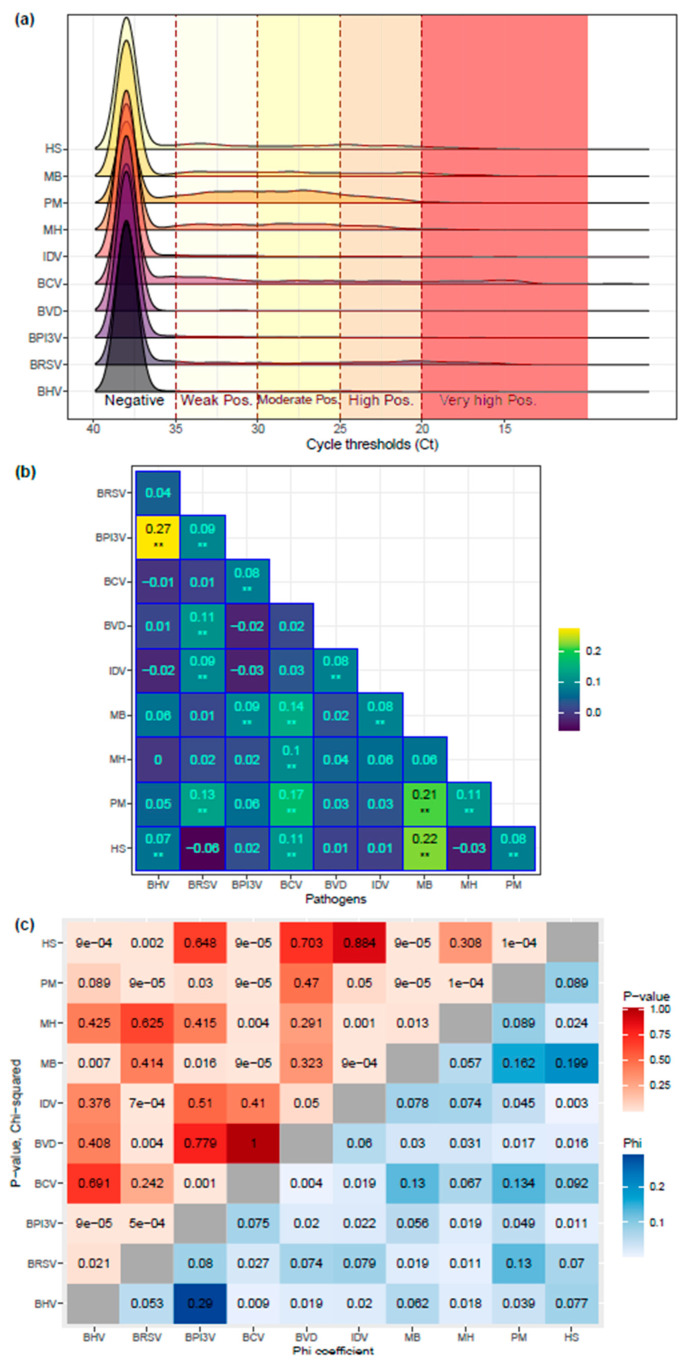
**Distribution of cycle thresholds for positivity signal** (**Ct**) **and correlation between Ct and pathogens** (**using Ct < 35 threshold**) **in 1875 respiratory samples using the 10-pathogen multiplex PCR method.** The density distribution of the Ct in all the tested samples to reach a positive signal is presented in panel (**a**), with interpretation in terms of positivity. The non-parametric Spearman rho correlation between pathogens and the raw Ct is presented in correlogram (**b**) (** indicating *p*-value < 0.005). Panel (**c**) shows a heatmap indicating the relationship between pathogen positivity (using Ct < 35 to define a positive sample and Ct ≥ 35 for defining a negative sample). The upper diagonal of the matrix indicates the *p*-value obtained from the Chi-squared method to look for correlation between pathogens (with Yates continuity correction when needed). The lower diagonal of the matrix indicates the strength of the correlation using the Phi coefficient. BHV1: bovine herpesvirus type 1; BRSV: bovine respiratory syncytial virus; BCV: bovine coronavirus; BPI3V: bovine parainfluenza virus type 3; BVDV: bovine viral diarrhea virus; IDV: influenza type D virus; MB: *Mycoplasmopsis bovis*; MH: *Mannheimia haemolytica*; PM: *Pasteurella multocida*; and HS: *Histophilus somni*.

**Figure 4 vetsci-11-00631-f004:**
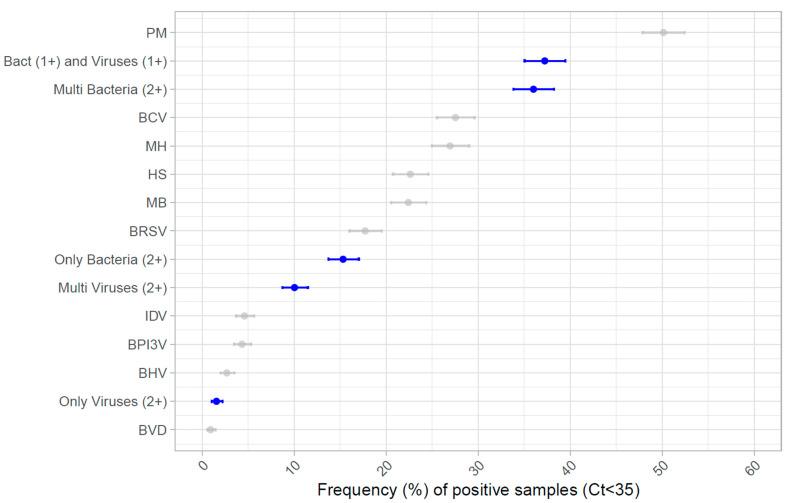
**Frequency of the different pathogens and pathogen association found during a 5-year period.** In grey, the frequency of individual pathogens is represented, alongside the 95% confidence intervals calculated with the Clopper–Pearson (exact) method, whereas the frequency of samples with only viruses (only viruses (2+), with 2 or more viruses and negative for bacteria), only bacteria (only bacteria (2+), with 2 or more bacteria and negative for viruses), multiple viruses (multi viruses (2+), with or without bacteria), multiple bacteria (multi bacteria (2+), with 2 or more bacteria and with or without viruses), and virus + bacteria association (Bact (1+) and Viruses (1+), for all samples with at least 1 virus and 1 bacterium) is highlighted in blue. *See abbreviations used in the legend under*
Figure 3.

**Figure 5 vetsci-11-00631-f005:**
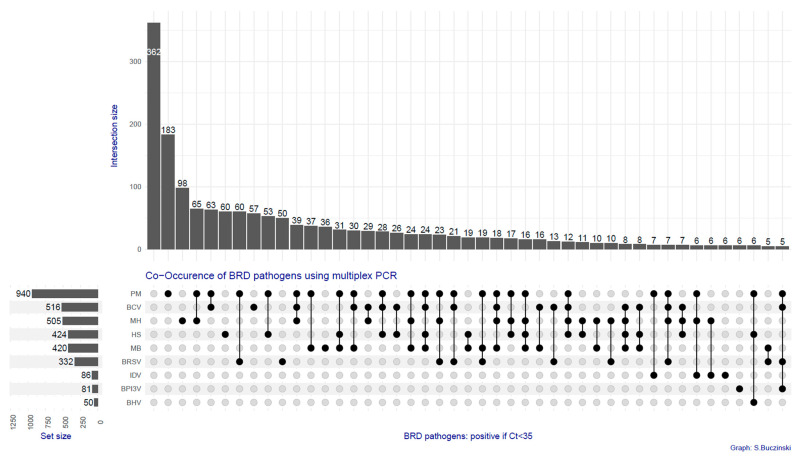
**Upset plot of the different combinations of pathogens found in 1875 bovine samples submitted for a 10-bovine-respiratory-pathogen multiplex PCR.** The 45 most common combinations found are presented. The bottom left horizontal bars represent the total count of pathogens in the dataset (BVDV is not presented since it is not associated with the 45 most common profiles). The black dots indicate positive samples for a particular agent. The vertical black line at the bottom links the black dots and represents a unique association of different pathogen combinations. The number of samples representing a particular association of positive and negative BRD agent profiles is indicated in the grey vertical bars, in descending order. *See abbreviations used in the legend under*
Figure 3.

**Table 1 vetsci-11-00631-t001:** Proportion and frequency of respiratory pathogens in 1875 samples submitted across the Province of Quebec, Canada, from 2019 to 2023.

		Viruses						Bacteria			
PCR Threshold ^1^	Interpretation	BHV	BRSV	PI3	BCV	BVDV	IDV	MB	MH	PM	HS
35+	Negative	1825 (97.3%)	1543 (82.3%)	1794 (95.7%)	1359 (72.5%)	1858 (99.1%)	1789 (95.4%)	1455 (77.6%)	1370 (73.1%)	935 (49.9%)	1451 (77.4%)
30–35	Weakly positive	13 (0.7%)	77 (4.1%)	40 (2.1%)	181 (9.7%)	11 (0.6%)	43 (2.3%)	142 (7.6%)	183 (9.8%)	364 (19.4%)	145 (7.7%)
25–30	Moderately positive	18 (1.0%)	62 (3.3%)	22 (1.2%)	102 (5.4%)	4 (0.2%)	19 (1.0%)	125 (6.7%)	201 (10.7%)	390 (20.8%)	112 (6.0%)
20–25	Highly positive	12 (0.6%)	104 (5.5%)	11 (0.6%)	93 (5.0%)	0 (0.0%)	16 (0.9%)	105 (5.6%)	104 (5.5%)	168 (9.0%)	123 (6.6%)
<20	Very highly positive	7 (0.4%)	89 (4.7%)	8 (0.4%)	140 (7.5%)	2 (0.1%)	8 (0.4%)	48 (2.6%)	17 (0.9%)	18 (1.0%)	44 (2.3%)
Cumulative proportion of positive (%) ^2^		**2.7%**	**17.7%**	**4.3%**	**27.5%**	**0.9%**	**4.6%**	**22.4%**	**26.9%**	**50.1%**	**22.6%**

^1^ Represents the number of cycle thresholds to obtain a positive fluorescence signal. ^2^ The sums of percentages do not necessarily add up exactly to 100% due to rounding issues. BHV1: bovine herpesvirus type 1; BRSV: bovine respiratory syncytial virus; BCV: bovine coronavirus; BPI3V: bovine parainfluenza virus type 3; BVDV: bovine viral diarrhea virus; IDV: influenza type D virus; MB: *Mycoplasmopsis bovis*; MH: *Mannheimia haemolytica*; PM: *Pasteurella multocida*; and HS: *Histophilus somni*.

**Table 2 vetsci-11-00631-t002:** Proportion and frequency of isolation of respiratory pathogens based on year, season, and number of individuals in the respiratory sample.

			Viruses						Bacteria			
Variable	Categories	Total Number of Samples ^1^	BHV1	BRSV	BPI3V	BCV	BVDV	IDV	MB	MH	PM	HS
**Year**	**2019**	318	17 (5.3%)	46 (14%)	16 (5.0%)	70 (22%)	6 (1.9%)	18 (5.7%)	83 (26%)	77 (24%)	152 (48%)	79 (25%)
	**2020**	343	8 (2.3%)	57 (17%)	13 (3.8%)	89 (26%)	4 (1.2%)	14 (4.1%)	92 (27%)	80 (23%)	184 (54%)	77 (22%)
	**2021**	349	5 (1.4%)	72 (21%)	14 (4.0%)	92 (26%)	1 (0.3%)	13 (3.7%)	65 (19%)	95 (27%)	179 (51%)	68 (19%)
	**2022**	437	11 (2.5%)	70 (16%)	18 (4.1%)	126 (29%)	5 (1.1%)	28 (6.4%)	99 (23%)	141 (32%)	225 (51%)	96 (22%)
	**2023**	413	8 (1.9%)	86 (21%)	19 (4.6%)	133 (32%)	1 (0.2%)	13 (3.1%)	75 (18%)	105 (25%)	187 (45%)	101 (24%)
	** *p-Values ^2^* **	*-*	*0.017*	*0.085*	*0.93*	*0.034*	*0.11*	*0.14*	*0.009*	*0.036*	*0.2*	*0.4*
**Season**	**Fall**	482	12 (2.5%)	67 (14%)	22 (4.6%)	125 (26%)	3 (0.6%)	10 (2.1%)	100 (21%)	133 (28%)	228 (47%)	116 (24%)
	**Winter**	579	14 (2.4%)	146 (25%)	31 (5.4%)	217 (37%)	7 (1.2%)	54 (9.3%)	157 (27%)	170 (29%)	318 (55%)	92 (20%)
	**Spring**	457	13 (2.8%)	81 (18%)	17 (3.7%)	112 (25%)	4 (0.9%)	15 (3.3%)	98 (21%)	124 (27%)	233 (51%)	68 (20%)
	**Summer**	342	10 (2.9%	37 (11%)	10 (2.9%)	56 (16%)	3 (0.9%)	7 (2.0%)	59 (17%)	71 (21%)	148 (43%)	145 (25%)
	** *p-Values ^2^* **	*-*	*0.95*	** *<0.001* **	*0.3*	** *<0.001* **	*0.8*	** *<0.001* **	** *0.003* **	*0.037*	** *0.004* **	*0.13*
**Pool**	**One individual**	1547	43 (2.8%)	253 (16%)	63 (4.1%)	392 (25%)	15 (1%)	66 (4.3%)	337 (22%)	381 (25%)	750 (48%)	356 (23%)
	**Two individuals**	159	1 (0.6%)	39 (25%)	4 (2.5%)	60 (38%)	1 (0.6%)	11 (6.9%)	35 (22%)	61 (38%)	82 (52%)	28 (18%)
	**Three and more**	154	5 (3.2%)	39 (25%)	13 (8.4%)	58 (38%)	1 (0.6)	9 (5.8%)	42 (27%)	56 (36%)	95 (62%)	37 (24%)
	** *p-Values* ** ** ^2^ **	*-*	*0.25*	** *<0.001* **	*0.02*	** *<0.001* **	*1*	*0.2*	*0.3*	** *<0.001* **	*0.007*	*0.3*

^1^ Fifteen samples (0.8%) are not included in this table due to absence of information on the number of individuals present in the pool. ^2^ The default significant level for multiple comparison is 0.005 to limit type 1 errors. See abbreviations used in the legend under Table 2.

**Table 3 vetsci-11-00631-t003:** Multivariable Poisson regression results associated with the number of respiratory pathogens isolated per submitted sample (*n* = 1860 ^1^).

		Number of Pathogens			Number of Bacteria			Number of Viruses		
		IRR	95%CI	*p*-Value ^2^	IRR	95%CI	*p*-Value	IRR	95%CI	*p*-Value
	Intercept	1.63	1.52–1.75	*<0.001*	1.16	1.07–1.27	*<0.001*	0.47	0.41–0.53	*<0.001*
**Type of sample**	Individual	Referent								
	Pool of 2	1.15	1.02–1.29	*0.02*	1.08	0.93–1.24	0.295	1.29	1.06–1.56	*0.010*
	Pool of 3+	1.32	1.18–1.47	*<0.001*	1.25	1.09–1.43	*0.001*	1.47	1.21–1.77	*<0.001*
**Season**	Fall	Referent								
	Winter	1.28	1.17–1.40	*<0.001*	1.14	1.02–1.27	*0.018*	1.63	1.39–1.90	*<0.001*
	Spring	1.02	0.92–1.12	0.759	1.00	0.89–1.12	0.947	1.06	0.89–1.27	0.508
	Summer	0.82	0.73–0.92	*0.001*	0.82	0.75	*0.02*	0.74	0.59–0.92	*0.007*

^1^ Fifteen samples are not included due to the absence of information concerning the number of individuals present in the pool. ^2^ The significance threshold is set to *p* < 0.05. IRR: incidence rate ratio; and CI: confidence interval.

## Data Availability

Data are not available.

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
