# Peer review of "Frequency of Bovine Respiratory Disease Complex Bacterial and Viral Agents Using Multiplex Real-Time qPCR in Quebec, Canada, from 2019 to 2023"

_vetsci, 2024, doi:10.3390/vetsci11120631_

Round 1

Reviewer 1 Report

Comments and Suggestions for Authors

This manuscript presents a retrospective analysis of bovine samples tested for respiratory pathogens in a diagnostic laboratory. The findings are centered around disease positivity rate and associations with co-infections, season, and pooling. The nature of the data set limits further analysis that would lead to novel insights. Overall, the basic findings have value but the manuscript requires revision for clarity as well as correcting numbers presented so they are consistent between text, figures, and tables. A number of specific points to address are provided below, but the entire manuscript should be revised.

1.     Consider revising the title to reflect the findings being presented; although qPCR was used in testing, disease incidence and associations are the focus of tables and figures.

2.     Correct and clearly explain sample numbers and percentages throughout text, tables, and figures. A total of 1,875 samples is written but different values are obtained when adding numbers presented. A few examples are:

a.      Figure 1 panel A adds up to 1,873, which also doesn’t match numbers on page 4 lines 172-173. Panel B adds up to 1545, text on page 5 line 188 states 1548.

b.     Page 9 lines 15 – positive for multiple virus (n = 188) and then on line 16 it states multiple viruses n=29. I really tried to understand this but I couldn’t tell where these numbers came from. Also, multiple viruses (n=188) + multiple bacteria (n=675) + bacteria and virus (n=698) + no pathogen detected (n=362) totals 1923, so these can’t be correct.

c.      Table 2 pool of 3 and more = 154, page 1- line 45 “153 samples contained 3 or more samples”.

d.     BCV 27.7% on page 13 line 129 versus 27.5% on Table 1

3.     Revise sample numbers on Page 1 lines 27-29. The way it is presented is very unclear, give actual number rather than “83.2% of 1860”, which isn’t a whole number.

4.     How many qPCR reactions were used to obtain results from these 10 pathogens? Page 1 line 14 states two assays, the methods describe 4 assays (page 3 lines 118-123), and other places suggest a single assay (page 12 line 87, Figure 5 title, and others). Please clarify.

5.     Lines 183-186 on page 5 should be part of Figure 1 legend.

6.     The placement of the sentence about individual samples on page 5 lines 187-188 is confusing, it should be moved to the previous paragraph.

7.     Figure 2 doesn’t add much value to the manuscript as presented. It is difficult to distinguish the shades of blue when the spaces are small. Because this same data is presented in other figures, I recommend deleting this figure as presented.

8.     Please be consistent with Influenza D virus abbreviation – both IDV and IVD are used.

9.     The Figure 3 legend is very confusing and needs revision. Figure 3b doesn’t contribute to the paper as the strongest finding is in the “weak” category, this could be replaced with a brief sentence. If the authors feel this is an important figure, I would consider moving it to supplemental data.

10.  The Figure 4 y-axis title “Agent” is included on the axis, please fix.

11.  Figure 5 – I think this is the best attempt made at presenting the co-infections but it isn’t clear. There is no difference between the color of black and grey bars, it would be better to choose colors that differ. The title implies the figure presents results of a single 10 pathogen multiplex PCR, and ‘combinations of pathogens’. Given that title, it is confusing to include bars for no pathogen and single pathogen results. Is the y-axis supposed to show the number of results? That makes sense for no pathogen (n=362) but not for PM only, which should be 940. This figure needs a better explanation or different presentation.

12.  Figure 6 – revise legend, select more distinctive symbols or use colors to distinguish results from other studies, add location to Wisselink reference in legend, and add numbers to the references in the legend so they can be found easily in the reference section.

13.  Page 13 line 146 refers to “the investigated outbreak” but there is no explanation or context for this. Please revise.

14.  Page 14 lines 196-203 is missing at least one reference, please add.

15.  There is no discussion of the fact that multiple samples were likely to come from a single farm/location, rather, samples seem to be treated as if they were individual. Given the infectious nature of these pathogens, positive results coming from the sample location would be expected and may alter statistical methods and/or interpretations.

16.  The conclusion requires revision for clarity. Additionally, this study presented the occurrence of multiple pathogens, not the importance.

17.  The Acknowledgments section on page 16 has template text, revise or delete as appropriate.

18.  Fix typographical errors throughout – a few examples are on page 2 line 59, page 14 lines 161 and 191.

19.  Either use continuous line numbering for the entire manuscript for start over on every page, don’t use both approaches.

Comments on the Quality of English Language

English language use needs revision throughout text and figure legends.

Author Response

Dear Editor,

We thank you and the reviewers for the critical and constructive review of our manuscript. We have tried to address each reviewer’s comment in the following sections. We have highlighted the changes throughout the revised manuscript in yellow to fasten the reviewing process. We also made a thorough revision of the manuscript sending it to a native English speaker to answer to the reviewers’ concerns concerning the English language. We sincerely hope that this revised version of the manuscript is now acceptable for publication.

This manuscript presents a retrospective analysis of bovine samples tested for respiratory pathogens in a diagnostic laboratory. The findings are centered around disease positivity rate and associations with co-infections, season, and pooling. The nature of the data set limits further analysis that would lead to novel insights. Overall, the basic findings have value but the manuscript requires revision for clarity as well as correcting numbers presented so they are consistent between text, figures, and tables. A number of specific points to address are provided below, but the entire manuscript should be revised.

Authors (AU): we thank the reviewer for his/her critical and constructive review of our manuscript. We are pleased to submit to you a revised version of the manuscript. All changes from the initial version of the manuscript have been highlighted in yellow to make the review process easier. We have addressed below each reviewer’s comment. We also double checked the numbers and we are sorry for the small discrepancies that were noted in the manuscript during the first review. We have corrected this point. We sincerely hope that our revised version of the manuscript would now be acceptable for publication.

  1. Consider revising the title to reflect the findings being presented; although qPCR was used in testing, disease incidence and associations are the focus of tables and figures.

AU: The title has been changed as suggested focussing on the main aim of the study which was to determine the percentage of positive samples (prevalence) of the main BRD agents. The title would now be read as : Prevalence of bovine respiratory disease complex bacterial and viral agents using multiplex real-time qPCR in Québec, Canada from 2019 to 2023. (in L 2-5)

  1. Correct and clearly explain sample numbers and percentages throughout text, tables, and figures. A total of 1,875 samples is written but different values are obtained when adding numbers presented. A few examples are:

AU: we have made a thorough review and reading of the numbers. As stated in the manuscript we had a total of 1,875 samples where we could assess the pathogens prevalence. However, we only had 1,860 samples where information was available concerning the number of individual animals pooled in a particular sample. 15 samples were indicated as pool with no indication of individual numbers within the pool. We have double checked all the data and redo the analyses for a final confirmation.

  1. Figure 1 panel A adds up to 1,873, which also doesn’t match numbers on page 4 lines 172-173. Panel B adds up to 1545, text on page 5 line 188 states 1548.

AU: We thank the reviewer for noticing this small discrepancy. There was a slight discrepancy (due to a previous R recoding line which was not updated in the submitted file) between the figure in the document vs Figure 1 pdf file (1 sample more in 2019 and 1 more in 2022). We re-ran all analyses to clarify and double check all numbers. We added the revised version of the figure 1 in the manuscript. The reviewer can now see that the panel A and B adds up to 1,875 (319+343+352+441+420=1,875).

The number of samples submitted by seasons were also rechecked for panel A.:

Summer: 65+50+54+96+79=344, 344/1875=18.35%

Spring: 85+101+78+102+96=462, 462/1875=24.64%

Winter: 76+116+136+119+136=583, 583/1875=31.09%

Fall: 93+76+84+124+109=486, 486/1875, 25.92%

For Panel B we have 328+1547= 1875 samples

Pool: 28+45+68+99+88=328 (328/1875=17.49%)

Individual sample :291+298+284+342+332=1547 (1547/1875=82.51%)

We are very sorry for this small mistake that was associated with a small block of coding. The problem is addressed now in the revised version of the manuscript.

  1. Page 9 lines 15 – positive for multiple virus (n = 188) and then on line 16 it states multiple viruses n=29. I really tried to understand this but I couldn’t tell where these numbers came from. Also, multiple viruses (n=188) + multiple bacteria (n=675) + bacteria and virus (n=698) + no pathogen detected (n=362) totals 1923, so these can’t be correct.

AU: We are sorry for this misunderstanding. Regarding this section we a priori distinguished 5 types of findings based on the results of the samples:

  • Proportion of sample positive for 1 particular agent (as summarized in Table 1)
  • Total number of agents present (which included either bacteria and/or virus in Figure 2, which also allows by deduction to know the proportion of samples with 1 or more viruses: for example: Figure 2: in the bar representing the samples with 2 pathogens (x-axis) found, we can see that roughly 40% of samples had 2 bacteria (so no virus), 45% of samples had 1 bacteria (and therefore 1 virus isolated) and the balance is the small proportion of samples with no bacteria found but (by deduction) 2 viruses (since the total number of pathogens was 2 in this bar). This is the main information given by this Figure 2 which we think is very interesting to have an idea of the balance between bacteria and viruses signals in the samples based on the total number of pathogens found. We have changed the colors of this figure to make it easier to read.
  • Total number of bacteria vs total number of virus present. (which counts the number of virus and bacteria). The proportion of samples with multiple viruses contains both samples with only 2+ viruses (also presented in Figure 4 with the label: “only viruses (2+) and samples with 2+ virus and at least 1 bacterium. The sum of both is equal to the label “Multi Viruses (2+)” in Figure 4. The same reasoning was applied for the bacteria (Multi Bacteria (2+) is corresponding to samples with only 2+ bacteria but no virus (Only bacteria (2+) and all samples with 2 or more bacteria and 1 or more virus).

Finally, the category bacteria and virus regrouped all samples with at least 1 virus and 1 bacterium (Bact (1+) and Viruses (1+) in figure 4. 

  • We specifically reported the proportion of samples with only 2 or more viruses (no bacteria: category (Only viruses (2+) and proportion of samples with only 2 or more bacteria (Only Bacteria (2+)) because we sought it was a pertinent finding do add for the reader.

We sincerely hope that this is easier to understand now. We have updated the materials and methods section (L168-173) of the manuscript and the legend in Figure 4 for clearly defining these different categories (L286-294).

  1. Table 2 pool of 3 and more = 154, page 1- line 45 “153 samples contained 3 or more samples”.

AU: the right number is 154. This typo was corrected throughout the manuscript.

  1. BCV 27.7% on page 13 line 129 versus 27.5% on Table 1

AU: sorry for this mistake. The prevalence of BCV was 27.5% is now indicated throughout the manuscript.

  1. Revise sample numbers on Page 1 lines 27-29. The way it is presented is very unclear, give actual number rather than “83.2% of 1860”, which isn’t a whole number.

AU: We are sorry for this, unfortunately this was a rounding issue. A total of 1547 samples were individual samples which is 83.17% of the 1860 where information was available. We have rounded the percentage to .01 in this part.

  1. How many qPCR reactions were used to obtain results from these 10 pathogens? Page 1 line 14 states two assays, the methods describe 4 assays (page 3 lines 118-123), and other places suggest a single assay (page 12 line 87, Figure 5 title, and others). Please clarify.

AU: We have clarified these details in the revised version of the manuscript (in L 120-130). Two kits were used detecting simultaneously the 4 bacteria under investigation Bovichek® MRB bacteria and the 4 main BRD viruses (BHV1, BRSV, BPI3V and BCV) using Bovichek® MRB virus qPCR kit. The IVD and BVDV were assessed using another multiplex kit which uses the IDV primers reported by Hause et al., 2013 and then by Saegerman et al., 2022. The BVDV was detected using the same test than described in Savard et al., (2022ab).

Saegerman, C., Gaudino, M., Savard, C., Broes, A., Ariel, O., Meyer, G. and Ducatez, F. Influenza D Virus in Respiratory Disease in Canadian, Province of Québec, Cattle: Relative Importance and Evidence of New Reassortment between Different Clades. Transbound Emerg. Dis. 2022, 69, 1227-1245. 10.1111/tbed.14085.

Hause BM, Ducatez M, Collin EA, Ran Z, Liu R, Sheng Z, Armien A, Kaplan B, Chakravarty S, Hoppe AD, Webby RJ, Simonson RR, Li F. Isolation of a novel swine influenza virus from Oklahoma in 2011 which is distantly related to human influenza C viruses. PLoS Pathog. 2013 Feb;9(2):e1003176. doi: 10.1371/journal.ppat.1003176. Epub 2013 Feb 7. PMID: 23408893; PMCID: PMC3567177.

Savard C, Provost C, Ariel O, Morin S, Fredrickson R, Gagnon CA, Broes A, Wang L.(2022) First report and genomic characterization of a bovine-like coronavirus causing enteric infection in an odd-toed non-ruminant species (Indonesian tapir, Acrocodia indica) during an outbreak of winter dysentery in a zoo. Transbound Emerg Dis 2022 Sep;69(5):3056-3065. doi: 10.1111/tbed.14300. Epub 2021 Aug 31.

Savard C, Ariel O, Fredrickson R, Wang L, Broes A. (2022) Detection and genome characterization of bovine kobuvirus (BKV) in faecal samples from diarrhoeic calves in Quebec, Canada. Transbound Emerg Dis. 2022 May;69(3):1649-1655. doi: 10.1111/tbed.14086. Epub 2021 Apr 20.

We have added the monography of the tests that we can include as supplemental files if needed

  1. Lines 183-186 on page 5 should be part of Figure 1 legend.

AU : Changed as suggested in L 197-200.

  1. The placement of the sentence about individual samples on page 5 lines 187-188 is confusing, it should be moved to the previous paragraph.

AU: Moved as suggested in L192-193.

  1. Figure 2 doesn’t add much value to the manuscript as presented. It is difficult to distinguish the shades of blue when the spaces are small. Because this same data is presented in other figures, I recommend deleting this figure as presented.

AU: We thank the reviewer for this comment. As previously indicated in our answer to the comment 2.b we think that this figure is interesting because giving complementary information to the table 1 and figure 4 and 5. This figure gives “at glance” the number of bacteria and viruses based on the total number of pathogens positively identified. We think that part of the confusion during initial review came from an inadequate choice of color that was associated with reading difficulties. We have changed the color palette in the revised version of the manuscript and we think this revised version can be easier to interpret

  1. Please be consistent with Influenza D virus abbreviation – both IDV and IVD are used.

AU: we are sorry for these inconsistencies. We have corrected this abbreviation (IDV) throughout the manuscript as suggested.

  1. The Figure 3 legend is very confusing and needs revision. Figure 3b doesn’t contribute to the paper as the strongest finding is in the “weak” category, this could be replaced with a brief sentence. If the authors feel this is an important figure, I would consider moving it to supplemental data.

AU : Thank you for raising these points. We think that it is important to note that this figure gives interesting information on the density of the Ct we observed for the different pathogens as well as the raw correlation and the dichotomous association (using a Ct<35 threshold) between pathogens using a matrix of correlation. We tried to follow the suggestions from the 2nd reviewer trying to improve the legend description for clarity and using information on the complementary approach in the Materials and method section (L156-157). We hope that the reviewer has a better understanding of this figure now.

  1. The Figure 4 y-axis title “Agent” is included on the axis, please fix.

AU : removed as suggested

  1. Figure 5 – I think this is the best attempt made at presenting the co-infections but it isn’t clear. There is no difference between the color of black and grey bars, it would be better to choose colors that differ. The title implies the figure presents results of a single 10 pathogen multiplex PCR, and ‘combinations of pathogens’. Given that title, it is confusing to include bars for no pathogen and single pathogen results. Is the y-axis supposed to show the number of results? That makes sense for no pathogen (n=362) but not for PM only, which should be 940. This figure needs a better explanation or different presentation.

AU: We thank you for noticing this problematic figure. The upset plot figure is trying to give at glance the most common combinations of tests results. This plot is composed of 3 parts: the left horizontal bars at the bottom left summarized the results of positive samples per pathogen. The bottom right dots indicate the combination of positive finding (black dots vs gray=negative findings). The vertical black line links the positive findings of a particular combination test result.

The upper part of the graph is a bar chart. The bar does not represent a particular pathogen but a particular association with different pathogens combination based in a descending order (from the most common to the less common). For example, the second vertical bar of the figure for PM (Pasteurella multocida) only indicates samples where only PM was found (n=183, no other pathogens found than PM). The total number of PM found in the dataset is indicated by the left horizontal bar (940 positive samples on 1875 samples). This means that the majority of samples where PM was found (940-183=747) were also associated with other pahogens findings. For example, the 4th bar indicate samples where Pasteurella multocida was found with Mannheimia haemolytica only in n=65 samples. The upset plot is therefore the very interesting method to see, at glance, in a descending order, the different associations of either clinical signs or findings where various combinations are possible (as in medical field as summarized by Blum and Gelfman, 2023). We added more details on the upset plot interpretation in the figure legend in L 303-310.

Concerning the colors that were chosen, we removed the extra colors added and kept the default color of the figure which is more contrasted and easier to understand.   

Blum, M., & Gelfman, L. P. (2023). Visualizing Multimorbidity in Chronically Ill Populations Using UpSet Plots. Journal of pain and symptom management, 65(4), e397-e398.

  1. Figure 6 – revise legend, select more distinctive symbols or use colors to distinguish results from other studies, add location to Wisselink reference in legend, and add numbers to the references in the legend so they can be found easily in the reference section.

AU: We modified the plot (new figure 6) as suggested and added the Wisselink study location as requested (L412).

  1. Page 13 line 146 refers to “the investigated outbreak” but there is no explanation or context for this. Please revise.

AU: we are sorry for this confusion. What we meant here was that:  when investigating a respiratory outbreak and finding BRSV positive animals, the virus is implicated in the outbreak since it is not found in healthy animals (except shortly after modified live virus vaccination). We have modified this section accordingly (in L 432)

  1. Page 14 lines 196-203 is missing at least one reference, please add.

AU : added as suggested

  1. There is no discussion of the fact that multiple samples were likely to come from a single farm/location, rather, samples seem to be treated as if they were individual. Given the infectious nature of these pathogens, positive results coming from the sample location would be expected and may alter statistical methods and/or interpretations.

AU : Unfortunately we do not have all information concerning the identification of the farm where the samples were taken. However, when looking for the data, the identification of the request could be an imperfect approximation of the site. The 1875 samples represented 1464 different requests with a median and IQR of 1 (1-1) samples per request. Therefore, the aggregation at the cluster level is supposed to be limited. We can add this extra information to the discussion if needed. However, one should keep in mind that this proxy is imperfect.

  1. The conclusion requires revision for clarity. Additionally, this study presented the occurrence of multiple pathogens, not the importance.

AU: we have modified the conclusion according to the reviewer’s suggestion (L532-540)

  1. The Acknowledgments section on page 16 has template text, revise or delete as appropriate.

AU: deleted as suggested

  1. Fix typographical errors throughout – a few examples are on page 2 line 59, page 14 lines 161 and 191.

AU: The manuscript has been sent to a native English reader for revision of the typo.

  1. Either use continuous line numbering for the entire manuscript for start over on every page, don’t use both approaches.

AU: we are sorry for this problem. The original submitted file was continuously numbered. Something happens and I don’t understand why. We hope we’ve fixed this issue.

Comments on the Quality of English Language English language use needs revision throughout text and figure legends.

AU: the manuscript has been sent for a thorough English revision by a native English speaker.

Reviewer 2 Report

Comments and Suggestions for Authors

This study describes the detection of 10 different bovine respiratory infectious agents by real-time qPCR from a single laboratory in Québec, Canada over a 5-year period. The paper includes valuable information about the relative frequency of BRD-pathogens in this area, but there are some major limitations that need to be addressed in order for the manuscript to be suitable for publication.

General considerations

The objective should be re-written to be more clearly related to a scientific question (abstract lines 23-24, and introduction lines 87-89). In the aim of the study might be showed that is an investigation of the relative frequency of 10-BRD associated pathogens in Québec, Canada, if the information about this subject is not available or scarce (showing this information in the introduction if this is the case). The hypotheses should also be reformulated to show more novelty, nevertheless the hypotheses could not be necessary in a descriptive study as this.

In my opinion, it could be more appropriate to use the term “frequency” of pathogens than prevalence. The samples have been submitted from a 5-year period, most of them or all, probably (but unknown) from BRD affected animals and therefore they could reflect the relative frequency of these pathogens among cases of BRD but not the proportion of the animal population “infected” with them.

The results should be more concise and precise. There is a lot of analysis of the data, but the relevant or novel findings are not clearly presented. Some of the figures are redundant.

The discussion should focus on the relevant results of these study.

Specific comments about issues that should be addressed or clarified

Line 25. “using two real-time qPCR multiplex assays” It seems that these state is not precise. In addition to these two assays, other two individual assays were used for influenza D virus and BVDV seem to be used.

Line 130-131. The interpretation of a positive test has not been clearly defined in the manuscript. How was the equation (line 133) used?

Line 151 What is the meaning of “the relationship between each bacterium and virus”?

Line 159 The significance threshold was a priori set at P<.005, which is perfectly admissible. Was the Bonferroni correction used?

Lines 172-178. This paragraph is a description of the samples used. In my opinion, these are not results. It would be better to include this in material and methods. Were variations of samples submitted by seasons and years statistically significant?

Lines 183-186. this paragraph would be better to be included in the footnote of figure 1.  

Line 187. Table 1 is mentioned in the text, but the relevant results are not described here.

Lines 187-188. “The majority of 187 submitted samples were individual samples as indicated in Figure 1 (82.52%, n=1548)”. This is reiterated. In addition, it should be included in material and methods.

Line 190 and figure 2. I think this figure does is redundant and does not apport new information.

Lines 193-199. Cts are semicuantitave results. What is the biological significance of the correlation of Cts between agents. Are they comparable? Please, explain and/or discuss this point.

Table 1. Section 3 is confusing. It should be eliminated for a clear presentation of the results. In addition, results of single samples or pools should be presented separately because this is a relevant information.

Lines 14-20. Likewise, the analysis of associations between the presence of multiple pathogens should be done separately for individual and pooled samples. Additionally, figure 4 seems to be redundant. The frequency of pathogens could be showed in table 1, whereas correlation of pathogens could be shown with figure 5.

Line 47-50. Possible differences between the results of single samples pooled samples was hypothesized in the objective the study. Therefore, this analysis should be performed and presented in a relevant way. However, in the text and the table is presented as one more of the analysis done.

Table 2. A different significance of < 0.005 was used? Why?

Figure 6 and the description of these results have been included in the discussion, but they are results. Nevertheless, this could be a help to prepare the comparative discussion of the results but unnecessary in the manuscript.

Line 86-87 respiratory disease would be more appropriate than pneumonia because not always pneumonia is present.

Line 87 current study

Lines 87-96 this is a mere description of the results.

Line 101 no definition of strong correlation is indicated by the authors. It is not clear the results that are being discussed. Line 104-106 seems to be speculative. Are there any relevant references?

Line 107-108 What is the meaning of “agents involved per se”. It is ambiguous.

Line 144 “keeping recent vaccination history in mind” it ambiguous. I suppose that is related with the possibility of false positive results in vaccinated animals with attenuated BRSV vaccine. Please, say it clearly.

Lines 160-162. The fact that 50% of the IVD positive samples were weak is an additional result (not presented in results). Anyway, it seems interesting, but it is not discussed. What could be the significance of the fact of this weak positive?

Line 168 “what was previously reported”. Please, included the references in the text.

Lines 172 “these percentages” It is confusing. Are the authors referring to their own results or the results of the UK study?

Line 182. “in Denholm, K., N. P. Evans, K. Baxter-Smith and P. Burr “, please, change by Denholm et al.

Lines 196-204. What is the reference of these results? What is the relevant of these results to discuss the results of this study?

Lines 211-212. “that the observed proportion of positive samples increased for BRSV, BCV and Mannheimia haemolytica” when using pooled samples is the result of this study and this should be the results discussed.  The previous discussion was a justification of the use of pooled samples, which is a justification of the methods used but not a discussion of results.

Lines 220-221 “Typical 220 BRSV and BCV outbreaks have been associated with cold conditions.” A reference is needed.

Lines 214-225. There are several references about the relationship between BRD pathogens shedding and season as O´Neil et al 2014 (https://doi.org/10.1136/vr.102574) and Pardon et al 2020 (https://doi.org/10.3168/jds.2019-17486). They could be included in the discussion.

Author Response

This study describes the detection of 10 different bovine respiratory infectious agents by real-time qPCR from a single laboratory in Québec, Canada over a 5-year period. The paper includes valuable information about the relative frequency of BRD-pathogens in this area, but there are some major limitations that need to be addressed in order for the manuscript to be suitable for publication.

Authors (AU): we thank the reviewer of his/her critical review of our manuscript. We have tried to answer to all questions and issues raised by the reviewer and have highlighted the changes of the manuscript in yellow to fasten the review process. We sincerely hope that our revised manuscript would now be acceptable for publication.

General considerations

The objective should be re-written to be more clearly related to a scientific question (abstract lines 23-24, and introduction lines 87-89). In the aim of the study might be showed that is an investigation of the relative frequency of 10-BRD associated pathogens in Québec, Canada, if the information about this subject is not available or scarce (showing this information in the introduction if this is the case). The hypotheses should also be reformulated to show more novelty, nevertheless the hypotheses could not be necessary in a descriptive study as this.

In my opinion, it could be more appropriate to use the term “frequency” of pathogens than prevalence. The samples have been submitted from a 5-year period, most of them or all, probably (but unknown) from BRD affected animals and therefore they could reflect the relative frequency of these pathogens among cases of BRD but not the proportion of the animal population “infected” with them.

The results should be more concise and precise. There is a lot of analysis of the data, but the relevant or novel findings are not clearly presented. Some of the figures are redundant.

The discussion should focus on the relevant results of these study.

AU: we thank the reviewer for these constructive comments. As suggested, we changed the introduction and objectives/hypotheses of the studies to be more descriptive and defining the association between the different pathogens found (L88-95). We also used the term “frequency” vs prevalence as suggested (L2-5) and throughout the text.

Specific comments about issues that should be addressed or clarified

Line 25. “using two real-time qPCR multiplex assays” It seems that these state is not precise. In addition to these two assays, other two individual assays were used for influenza D virus and BVDV seem to be used.

AU: we have modified this part accordingly in L 26 and L126-130

Line 130-131. The interpretation of a positive test has not been clearly defined in the manuscript. How was the equation (line 133) used?

AU: the interpretation was based the laboratory report which consider weakly positive tests for Ct≥35. These thresholds were used for all pathogens (L134). There is always a risk of categorization associated with oversimplification of a biologic process (eg: 35.1 and 34.9 Ct would be classified differently despite very small difference of Ct). However, we kept this categorization for its inherent simplicity and practical applicability. We also indicated (L142-144) that the interpretation of Ct is depending on laboratory-pathogen-reaction.

Line 151 What is the meaning of “the relationship between each bacterium and virus”?

AU: the relationship aims to detect association of positive findings for pairs of pathogens. For this part of the examination compared 2 by 2 tables of positive vs negative samples of pathogens pairs and look for presence of concomitant findings of positive or negative tests results using a Chi-squared test. We’ve added specific information in L157-158 and 159-160.

Line 159 The significance threshold was a priori set at P<.005, which is perfectly admissible. Was the Bonferroni correction used?

AU: as indicated (L168), it was selected a priori for this part. The Bonferroni correction, despite appealing because of its simplicity has the main limitations of being too much conservative when > 20 tests are used (small commentary in Perneger, 1998). We preferred to cope with a pragmatic approach as introduced by the work of John Ioannidis (2018)

Perneger TV. What's wrong with Bonferroni adjustments. BMJ. 1998 Apr 18;316(7139):1236-8. doi: 10.1136/bmj.316.7139.1236.

Ioannidis, J. P. The Proposal to Lower p Value Thresholds to .005. JAMA 2018, 319, 1429-30. 10.1001/jama.2018.1536.

Lines 172-178. This paragraph is a description of the samples used. In my opinion, these are not results. It would be better to include this in material and methods. Were variations of samples submitted by seasons and years statistically significant?

AU: We respectively disagree with the reviewer point of view. Since we are describing the samples characteristics in terms of seasons and type of samples as well as giving information on the number of samples (and proportion) for each category we consider these information as results. The other 2 reviewers of the manuscript did not mention that comment. For this reason we want to keep this paragraph as a result. This information was also  added in the results section of the O’Neil (2024) study which was suggested by the reviewer. 

As indicated in the manuscript we also looked for variation of samples between seasons and pooled vs individual samples in L192-193. Samples were most commonly sent in fall and winter and proportion of pooled samples increased over the years.

O'Neill R, Mooney J, Connaghan E, Furphy C, Graham DA. Patterns of detection of respiratory viruses in nasal swabs from calves in Ireland: a retrospective study. Vet Rec. 2014 Oct 11;175(14):351. doi: 10.1136/vr.102574.

Lines 183-186. this paragraph would be better to be included in the footnote of figure 1.  

AU: done as suggested in L199-202

Line 187. Table 1 is mentioned in the text, but the relevant results are not described here.

AU: This has been changed as suggested in L206-207

Lines 187-188. “The majority of 187 submitted samples were individual samples as indicated in Figure 1 (82.52%, n=1548)”. This is reiterated. In addition, it should be included in material and methods.

AU: see previous comments. This sentence has been moved in L194-195.

Line 190 and figure 2. I think this figure does is redundant and does not apport new information.

AU: we respectfully disagree with the reviewer opinion but we agree that we had not explained correctly the importance of this figure and the complementary information provided by this figure in the first version of our manuscript. In the current study, we a priori distinguished 5 types of complementary findings based on the results of the samples:

  • Proportion of sample positive for 1 particular agent (as summarized in Table 1)
  • Total number of agents present (which included either bacteria and/or virus in Figure 2, which also allows by deduction to know the proportion of samples with 1 or more viruses: for example: Figure 2: in the bar representing the samples with 2 pathogens (x-axis) found, we can see that roughly 40% of samples had 2 bacteria (so no virus), 45% of samples had 1 bacteria (and therefore 1 virus isolated) and the balance is the small proportion of samples with no bacteria found but (by deduction) 2 viruses (since the total number of pathogens was 2 in this bar). This is the main information given by this Figure 2 which we think is very interesting to have an idea of the balance between bacteria and viruses signals in the samples based on the total number of pathogens found. We have changed the colors of this figure to make it easier to read.
  • Total number of bacteria vs total number of virus present. (which counts the number of virus and bacteria). The proportion of samples with multiple viruses contains both samples with only 2+ viruses (also presented in Figure 4 with the label: “only viruses (2+) and samples with 2+ virus and at least 1 bacterium. The sum of both is equal to the label “Multi Viruses (2+)” in Figure 4. The same reasoning was applied for the bacteria (Multi Bacteria (2+) is corresponding to samples with only 2+ bacteria but no virus (Only bacteria (2+) and all samples with 2 or more bacteria and 1 or more virus). Finally, the category bacteria and virus regrouped all samples with at least 1 virus and 1 bacterium (Bact (1+) and Viruses (1+) in figure 4.
  • We specifically reported the proportion of samples with only 2 or more viruses (no bacteria: category (Only viruses (2+) and proportion of samples with only 2 or more bacteria (Only Bacteria (2+)) because we sought it was a pertinent finding do add for the reader.

We hope that we have adequately explained these points to the reviewer. We have added information on these aspects in L169-174, 264-273, 288-296, 303-312.

Lines 193-199. Cts are semicuantitave results. What is the biological significance of the correlation of Cts between agents. Are they comparable? Please, explain and/or discuss this point.

AU: Thanks for noting this point. Even if we cannot compare directly the meaning of Ct for different pathogens in terms of number of infectious agents in the sample, we can have interesting information from a semiquantitative result. Knowing if the samples with high value Ct of pathogens have also high Ct values of another pathogen would be clinically meaningful. We used a non-parametric Spearman test to look for Ct as a continuous non-parametric indicator and also looking for the association of pathogens positivity as indicated in the previous comment using Chi-squared tests. We’ve added specific information in L157-158 and 159-160.

Table 1. Section 3 is confusing. It should be eliminated for a clear presentation of the results. In addition, results of single samples or pools should be presented separately because this is a relevant information.

AU: we’ve removed section 3 as suggested. Concerning the pooled samples differentiation for pathogens results we think we have already address it in Table 2 (last section when we can see the frequency of individual BRD pathogens in pooled samples (bottom rows of table 2).

Lines 14-20. Likewise, the analysis of associations between the presence of multiple pathogens should be done separately for individual and pooled samples. Additionally, figure 4 seems to be redundant. The frequency of pathogens could be showed in table 1, whereas correlation of pathogens could be shown with figure 5.

AU: we thank the reviewer for this suggestion. Since we had a relatively wide variation of individual per pool and also due to the limited information we had from these samples we only looked for the presence of pathogens based on type of sample (individual vs pooled (2 vs 3+ animals) as indicated in table 2 as well as on the impact that the pool could have on the detected pathogens (Poisson Regression part). We think that the figure 4 and 5 are complementary. The figure 4 allows the reader to see the different viral bacterial association “at glance” vs the figure 5 which gives more detailed view of the pathogens’ association. I think that the suggestion of the reviewer would also necessitate to control for the number of samples within a pool and we may be limited by the distribution of individual samples within a pool in our dataset. 

Line 47-50. Possible differences between the results of single samples pooled samples was hypothesized in the objective the study. Therefore, this analysis should be performed and presented in a relevant way. However, in the text and the table is presented as one more of the analysis done.

Table 2. A different significance of < 0.005 was used? Why?

AU: For this objective, we used a regular multivariable generalized linear model (ie Poisson count regression). We used the standard P<.05 model due to the fact that the risk of inflated type 1 error was much lower than in other analyses (eg less variables to enter in the model vs multiple independent tests).

Figure 6 and the description of these results have been included in the discussion, but they are results. Nevertheless, this could be a help to prepare the comparative discussion of the results but unnecessary in the manuscript.

AU: We think that this figure is useful as stated by another reviewer but we are open to remove it if needed.

Line 86-87 respiratory disease would be more appropriate than pneumonia because not always pneumonia is present.

AU : changed as suggested

Line 87 current study

AU : changed as suggested

Lines 87-96 this is a mere description of the results.

AU : we opted for this small recapitulation of the main results since many different results are reported and we consider that these are the most clinically relevant findings that would also guide our discussion process.

Line 101 no definition of strong correlation is indicated by the authors. It is not clear the results that are being discussed. Line 104-106 seems to be speculative. Are there any relevant references?

AU : we based this sentence on the benchmarks to interpret Spearman rho correlation and Phi coefficient (L153-165) in the M and M section.

Line 107-108 What is the meaning of “agents involved per se”. It is ambiguous.

AU : we removed this term as suggested. We just wanted to indicate that the number of agents found in respiratory secretion may depend on the agent found (large amount of viruses vs smaller numbers of bacteria). This also comes with the correlation between the Ct number and number of infectious agents.

Line 144 “keeping recent vaccination history in mind” it ambiguous. I suppose that is related with the possibility of false positive results in vaccinated animals with attenuated BRSV vaccine. Please, say it clearly.

AU: Yes it is exactly what we had in mind (L433-435).

Lines 160-162. The fact that 50% of the IVD positive samples were weak is an additional result (not presented in results). Anyway, it seems interesting, but it is not discussed. What could be the significance of the fact of this weak positive?

AU: It is difficult to make any formal hypothesis in the absence of more information on the nature of samples submitted and its relation with the clinical state of the cattle. Maybe IDV is shed earlier in the course of a BRD episode and when the calves are detected as sick by the veterinarians a smaller shedding can be observed? We did not want to commit to much on that aspect but would be open to do it if the reviewer thinks it is very important.

Line 168 “what was previously reported”. Please, included the references in the text.

AU: added as suggested (L457)

Lines 172 “these percentages” It is confusing. Are the authors referring to their own results or the results of the UK study?

AU: It was the UK study. Modified in L 462

Line 182. “in Denholm, K., N. P. Evans, K. Baxter-Smith and P. Burr “, please, change by Denholm et al.

AU: changed as suggested (L471)

Lines 196-204. What is the reference of these results? What is the relevant of these results to discuss the results of this study?

AU: we added the reference. We think it is important to discuss these results because we do not have a lot of information on the impact of pooling respiratory samples on the final result of the pool and its interpretation. For example, if only 1 of the n individuals tested in a low shedder of 1 pathogen, how pooling could affect the results based on the initial shedding and number of negative individuals in the pool (ie giving a false negative results)? We agree that this is a broad question but we think this point of discussion is important to be addressed in the future. As added in L502-505. We also thank the reviewer for the suggestion of O’Neil study which has also been added in the revised manuscript.

Lines 211-212. “that the observed proportion of positive samples increased for BRSV, BCV and Mannheimia haemolytica” when using pooled samples is the result of this study and this should be the results discussed.  The previous discussion was a justification of the use of pooled samples, which is a justification of the methods used but not a discussion of results.

AU: we thank the reviewer for this comment. We elaborated a little more on that particular aspect in L501- also adding a specific reference on the pooling of respiratory samples for bacteria diagnosis who reported the impact of pooling of M.bovis and H.somni positive samples (Goecke et al., 2021)

Goecke NB, Nielsen BH, Petersen MB, Larsen LE. Design of a High-Throughput Real-Time PCR System for Detection of Bovine Respiratory and Enteric Pathogens. Front Vet Sci. 2021 Jun 24;8:677993. doi: 10.3389/fvets.2021.677993.

Lines 220-221 “Typical 220 BRSV and BCV outbreaks have been associated with cold conditions.” A reference is needed.

AU: added as suggested.

Lines 214-225. There are several references about the relationship between BRD pathogens shedding and season as O´Neil et al 2014 (https://doi.org/10.1136/vr.102574) and Pardon et al 2020 (https://doi.org/10.3168/jds.2019-17486). They could be included in the discussion.

AU: We thank the reviewer for providing these additional references. We have added them to the manuscript as well as another Spanish study showing the winter patterns of BRD with viral component (Calderon Bernal et al., 2023).

Calderón Bernal JM, Fernández A, Arnal JL, Baselga C, Benito Zuñiga A, Fernández-Garyzábal JF, Vela Alonso AI, Cid D. Cluster analysis of bovine respiratory disease (BRD)-associated pathogens shows the existence of two epidemiological patterns in BRD outbreaks. Vet Microbiol. 2023 May;280:109701. doi: 10.1016/j.vetmic.2023.109701.

Reviewer 3 Report

Comments and Suggestions for Authors

Comments:

This study by Buczinski et al. is competently written. The statistical analysis section is well described, which is appreciated. The discussion section is also well-written. The authors do a decent job of discussing the dataset's individual limitations. One of the most significant shortcomings of the study is the use of "pooled" samples. The accompanying "hypothesis" is also very weak and illogical. The authors address this in the discussion, which is a step in the right direction.

Nevertheless, it does not solve the issue that the pooled vs individual sample portion of the results can be overinterpreted. Given the nature of sample collection, there are so many variables, and the near complete lack of information regarding pooling methods makes it extremely difficult to interpret. These results and discussion should be removed entirely. Since the authors use a commercially available kit, details on the qPCR assay methodology are incredibly sparse, making it challenging to compare head-to-head raw Ct values and differences in "viral load." Overall, the study is informative and adds to existing literature on subjects similar to those originating from other countries. However, this is not a hypothesis-driven study, and the authors should not try to present it as such. 

Simple Summary

The summary should be reworked overall.

Line 16: The sentence "The pathogens findings…" is hard to interpret. Please rephrase.

Abstract

Line 21: Define the BRD acronym here. 

Line 25: Remove "..from a single laboratory in.." This detail is unnecessary.

Line 27: Restructure the sentence as "Majority of samples were collected from individuals (%) and the rest from pooled animals (2 (%) or ≥3 (%))".

Introduction

Line 59: "..nasal OR deep.."

Line 61: "..agreement BETWEEN tracheal AND bronchial.."

Line 81-85: Sentence hard to follow. Break up into multiple, short sentences.

Line 89-90: The nature of this study does not need to be hypothesis-driven. The authors present the rationale for designing a multiplex PCR well in the previous paragraph. The assay reports numbers comparable to other studies (Figure 6), which is the ultimate goal of the study. 

Multiple hypotheses are presented that are very weak, given the lack of sample information (as described in materials and methods). For example, the authors' hypothesis is "We also hypothesized that a difference between pooled vs individual sample results would be observed with an increased number of multiple BRD agents when using pooled samples." Pooling samples with different microbiome compositions would increase the microbiome diversity of the pooled sample. The outcome should be dependent on the heterogeneity across individuals. Whether this carries any biological significance depends on how the samples were pooled (see below).

Line 103: What does "pooled sample" mean? The authors should include information regarding how the pooled samples were generated, not just how many individuals are in each pool (Fig 1B). For instance, a pooled sampling of co-housed cattle from a single location, collected from the same site (nasal swab vs. fecal samples) at a single time point, differs from samples from cattle from other herds, collected at multiple times throughout the year being pooled together. 

Materials and Methods

Line 96: Do the authors mean the data was collected from the Biovet internal database? Please clarify here, as it has been in the COI section. 

Line 100: Check punctuation - "...filled by referring veterinarians, it was impossible.."

Lines 115-123: How does the kit work? Is it probe-based? Are standards included? The methodology is not transparent, given that this study is a product of a commercial entity, which makes it difficult to interpret. For example, Ct values can be used for clinical diagnostics in a binary positive/negative manner. Interpreting more is risky, given differences in sample prep and assay variability. However, it is possible to interpret viral loads from Ct values if the assay has been optimized and standardized. The authors provide no such data.

Most viruses are RNA; one is DNA, and the other is bacteria DNA. Is the qPCR assay one-step or two-step? How do the authors account for differences in sample prep that lead to differences in abundance?

As the authors discuss in lines 135-137, Ct values cannot be interpreted because of differences in laboratory variability. It is good that the authors address the limitations of their approach.   

Results

Line 194: The sentence structure is a bit confusing. The correlations are pretty weak, so the authors should avoid using "strongest correlations." Instead, use something like "Out of (number) pairwise comparisons, we found weak correlations between.." Looking at the Ct distribution (Table 1, Figure 3a), it is apparent that some of these correlations, like BHV and BPI3V, are based on low Ct values. Are most correlations driven by low Ct, i.e., the absence of these pathogens? In other words, what would the correlation look like if the authors only used Ct <25 thresholds?

Figure 1-3,5,6: Use consistent and larger (2X) font sizes for all labels. The Y-axis and X-axis labels are tough to see. 

Figure 3: Properly declare what each "value" means in the legends, i.e., P-value and Phi-coefficient. Adjust color transparency to ensure that overlaying numbers are correctly visible. 

Note: New line numbering starts on page 8.

Line 47: I am not sure how readers are supposed to interpret this result given the missing context of how samples were pooled (see above) 

Discussion

Line 87: The current studies study

Line 89: "..indicate that among viral components viruses.."

Line 109: "Bacterial agents were generally more frequently.." How much of this is due to the sample extraction process? Some viruses on the authors' list are RNA viruses, which are more susceptible to degradation. 

Comments on the Quality of English Language

A thorough grammar check is recommended.

Author Response

This study by Buczinski et al. is competently written. The statistical analysis section is well described, which is appreciated. The discussion section is also well-written. The authors do a decent job of discussing the dataset's individual limitations. One of the most significant shortcomings of the study is the use of "pooled" samples. The accompanying "hypothesis" is also very weak and illogical. The authors address this in the discussion, which is a step in the right direction.

Nevertheless, it does not solve the issue that the pooled vs individual sample portion of the results can be overinterpreted. Given the nature of sample collection, there are so many variables, and the near complete lack of information regarding pooling methods makes it extremely difficult to interpret. These results and discussion should be removed entirely. Since the authors use a commercially available kit, details on the qPCR assay methodology are incredibly sparse, making it challenging to compare head-to-head raw Ct values and differences in "viral load." Overall, the study is informative and adds to existing literature on subjects similar to those originating from other countries. However, this is not a hypothesis-driven study, and the authors should not try to present it as such.

Authors (AU): we thank the reviewer for his/her critical and constructive review of our manuscript. We are pleased to submit to you a revised version of the manuscript. All changes from the initial version of the manuscript have been highlighted in yellow to make the review process easier. We have addressed below each reviewer’s comment. Concerning the comments on the pooled samples, we respectfully disagree with removing the data from the analyses for various reasons:

  • First of all, this method is commonly used by field practitioners and has also been promoted as a way to decrease costs to investigate BRD outbreaks and previous comparable studies (Denholm et al., 2023).
  • The use of pooled samples was performed directly in the lab at the veterinarian request when submitting multiple samples. Therefore, we are convinced that the procedure to obtain the pooled samples is similar to Denholm study (2023). We have clarified this point in the revised version of the manuscript to clarify this point in L115-117. We thank the reviewer for pointing this point.
  • Despite the limitations due to the nature of the study design we think it is important to mention it and because of dilution associated with the processing of the sample our hypothesis was based on what was already found for covid19 studies especially when initial shedding is low at the individual level (Furstenau et al., 2020). We had this hypothesis in mind when analysing the pooled samples which was as also hypothesized from a thorough review of respiratory disease diagnosis by Pardon and Buczinski (2020).
  • We agree that this study has various limitations that are inherent to the nature of the data and that the hypothesis we had for this point was not the primary aim of the study (which was to describe the prevalence of the pathogens). We have reformulated this point in L92-95, and also emphasizing it on the discussion section in L499-502.

Pardon B, Buczinski S. Bovine Respiratory Disease Diagnosis: What Progress Has Been Made in Infectious Diagnosis? Vet Clin North Am Food Anim Pract. 2020 Jul;36(2):425-444. doi: 10.1016/j.cvfa.2020.03.005. PMID: 32451034; PMCID: PMC7244442.

Furstenau, T. N., Cocking, J. H., Hepp, C. M. and Fofanov, V. Y. Sample Pooling Methods for Efficient Pathogen Screening: Practical implications. PLoS One 2020, 15, e0236849. 10.1371/journal.pone.0236849.

We sincerely hope that our revised version of the manuscript would now be acceptable for publication.

Simple Summary

The summary should be reworked overall.

AU: we have tried to modify the summary as requested.

Line 16: The sentence "The pathogens findings…" is hard to interpret. Please rephrase.

AU: changed as suggested

Abstract

Line 21: Define the BRD acronym here.

AU: done as suggested.

Line 25: Remove "..from a single laboratory in.." This detail is unnecessary.

AU: done as suggested.

Line 27: Restructure the sentence as "Majority of samples were collected from individuals (%) and the rest from pooled animals (2 (%) or ≥3 (%))".

AU: Changed as suggested

Introduction

Line 59: "..nasal OR deep.."

AU: corrected as suggested

Line 61: "..agreement BETWEEN tracheal AND bronchial.."

AU: corrected. The sentence aimed to mention that DNS results agreed with tracheal and bronchial results.

Line 81-85: Sentence hard to follow. Break up into multiple, short sentences.

AU: We have modified this sentence as indicated.

Line 89-90: The nature of this study does not need to be hypothesis-driven. The authors present the rationale for designing a multiplex PCR well in the previous paragraph. The assay reports numbers comparable to other studies (Figure 6), which is the ultimate goal of the study.

Multiple hypotheses are presented that are very weak, given the lack of sample information (as described in materials and methods). For example, the authors' hypothesis is "We also hypothesized that a difference between pooled vs individual sample results would be observed with an increased number of multiple BRD agents when using pooled samples." Pooling samples with different microbiome compositions would increase the microbiome diversity of the pooled sample. The outcome should be dependent on the heterogeneity across individuals. Whether this carries any biological significance depends on how the samples were pooled (see below).

AU: The optimal pooling strategy for analyzing respiratory samples still needs to be determined. We have added information on the way the samples were pooled. Our main hypothesis was based on the general information concerning the use of pooled samples in cattle health. As noticed by the reviewer this was not the main objective of the study. However, we think it was important to look for difference between the 2 types of samples (individual vs pooled). However, the study design does not allow us to make any causal inference. We modified a little bit the section concerning the pooled samples removing the hypothesis (L115-117) and add a specific sentence in the discussion section (L499-502).

Line 103: What does "pooled sample" mean? The authors should include information regarding how the pooled samples were generated, not just how many individuals are in each pool (Fig 1B). For instance, a pooled sampling of co-housed cattle from a single location, collected from the same site (nasal swab vs. fecal samples) at a single time point, differs from samples from cattle from other herds, collected at multiple times throughout the year being pooled together.

AU: as previously described we have added information on the way the samples were pooled.  All pooled samples came from the same farms and the same types of samples were pooled as indicated in the text using a similar preparation. Various pooling strategies have been mentioned for assessment of different infectious diseases status in group of cattle (eg BVDV (ex: Smith et al., 2008 or Anaplasmosis and theileriosis more recently (Burgess et al., 2024)). However, little is known for pathogens involved in BRD outbreaks.

Burgess C, Todd SM, Hungerford L, Lahmers K. Determining diagnostic sensitivity loss limits for sample pooling in duplex rtPCR surveillance testing: Theileria orientalis and Anaplasma marginale. J Vet Diagn Invest. 2024 Oct 26:10406387241287516. doi: 10.1177/10406387241287516.

Smith RL, Sanderson MW, Walz PH, Givens MD. Sensitivity of polymerase chain reaction for detection of bovine viral diarrhea virus in pooled serum samples and use of pooled polymerase chain reaction to determine prevalence of bovine viral diarrhea virus in auction market cattle. J Vet Diagn Invest. 2008 Jan;20(1):75-8. doi: 10.1177/104063870802000115.

Materials and Methods

Line 96: Do the authors mean the data was collected from the Biovet internal database? Please clarify here, as it has been in the COI section.

AU : we have modified this section as suggested.

Line 100: Check punctuation - "...filled by referring veterinarians, it was impossible.."

AU : changed as suggested.

Lines 115-123: How does the kit work? Is it probe-based?

AU : Yes the kit is probe-based. We did not give too detailed information on these kits since they are commercial kits and have already been previously described in previous publications (Saegerman et al., 2022, Savard et al 2022). However, we can add particular information if needed.

Saegerman, C., Gaudino, M., Savard, C., Broes, A., Ariel, O., Meyer, G. and Ducatez, F. Influenza D Virus in Respiratory Disease in Canadian, Province of Québec, Cattle: Relative Importance and Evidence of New Reassortment between Different Clades. Transbound Emerg. Dis. 2022, 69, 1227-1245. 10.1111/tbed.14085.

Savard C, Provost C, Ariel O, Morin S, Fredrickson R, Gagnon CA, Broes A, Wang L.(2022) First report and genomic characterization of a bovine-like coronavirus causing enteric infection in an odd-toed non-ruminant species (Indonesian tapir, Acrocodia indica) during an outbreak of winter dysentery in a zoo. Transbound Emerg Dis 2022 Sep;69(5):3056-3065. doi: 10.1111/tbed.14300. Epub 2021 Aug 31.

Savard C, Ariel O, Fredrickson R, Wang L, Broes A. (2022) Detection and genome characterization of bovine kobuvirus (BKV) in faecal samples from diarrhoeic calves in Quebec, Canada. Transbound Emerg Dis. 2022 May;69(3):1649-1655. doi: 10.1111/tbed.14086. Epub 2021 Apr 20.

Are standards included?

AU : Yes the standard were included as added in L127.

The methodology is not transparent, given that this study is a product of a commercial entity, which makes it difficult to interpret. For example, Ct values can be used for clinical diagnostics in a binary positive/negative manner. Interpreting more is risky, given differences in sample prep and assay variability. However, it is possible to interpret viral loads from Ct values if the assay has been optimized and standardized. The authors provide no such data.

AU: We thank the reviewer for this comment. As indicated revised version of the manuscript, we did not attempt to give an estimate of the number of copies of genetic material in the sample. However, the Ct value is associated with the number of copies (viral or bacterial load) in the sample. We followed a similar framework as presented by Denholm et al (2023) which also reported Ct results in an ordinal way based on Ct results. Since the preparation of the samples were standardized we think that semi-quantitative interpretation of the results can be helpful. We have indicated this limitation in L500-502.

Denholm K, Evans NP, Baxter-Smith K, Burr P. Retrospective study of the relative frequency of cattle respiratory disease pathogens in clinical laboratory samples submitted by UK veterinary practices. Vet Rec. 2024 Sep 21;195(6):e4434. doi: 10.1002/vetr.4434.

We have added the monography of the tests that we can include as supplemental files if needed

Most viruses are RNA; one is DNA, and the other is bacteria DNA. Is the qPCR assay one-step or two-step?

AU: this was a 1 step.

 How do the authors account for differences in sample prep that lead to differences in abundance?

AU: As indicated, we used commercial extraction kits which are extracting all nucleic acids (DNA and RNA) we have no reason to think it could influence the results.

We have added the monography of the tests that we can include as supplemental files if needed

As the authors discuss in lines 135-137, Ct values cannot be interpreted because of differences in laboratory variability. It is good that the authors address the limitations of their approach.

AU: we have addressed this in the discussion section in L499-502.

Results

Line 194: The sentence structure is a bit confusing. The correlations are pretty weak, so the authors should avoid using "strongest correlations." Instead, use something like "Out of (number) pairwise comparisons, we found weak correlations between.." Looking at the Ct distribution (Table 1, Figure 3a), it is apparent that some of these correlations, like BHV and BPI3V, are based on low Ct values. Are most correlations driven by low Ct, i.e., the absence of these pathogens? In other words, what would the correlation look like if the authors only used Ct <25 thresholds?

AU: we thank the reviewer for this interesting comment. We agree that the Spearman rho correlation could not be extrapolated beyond the current dataset since it is representing a screenshot of available results and that for some pathogens that have a low frequency of isolation the variability of Ct is lower than others. This is why we think that this information is complementary to information using a dichotomous approach of the Ct at a Ct<35 level (positive threshold level). We have indicated this more properly in the materials and method section in L156-157. We can observe that the P-value is still statistically significant when using this dichotomous approach (P=.00009) but that the Phi coefficient is 0.29 denoting a small association (Figure 3c).

Figure 1-3,5,6: Use consistent and larger (2X) font sizes for all labels. The Y-axis and X-axis labels are tough to see.

AU: The size of the labels and axis scales have been increased as suggested. For the upset plot, we changed the colors for a better reading of the figure. We also improved the legend to make it easier to understand. We hope this is easier to follow now.

Figure 3: Properly declare what each "value" means in the legends, i.e., P-value and Phi-coefficient. Adjust color transparency to ensure that overlaying numbers are correctly visible.

AU: The figure and legends have been changed in order to improve the contrasts.

Note: New line numbering starts on page 8.

AU: We are sorry for this mistake, we don’t know what happened during the initial submission. 

Line 47: I am not sure how readers are supposed to interpret this result given the missing context of how samples were pooled (see above)

AU: We hope that the precision we gave earlier are clarifying this point.

Discussion

Line 87: The current studies study

AU: changed as requested

Line 89: "..indicate that among viral components viruses.."

AU: changed as suggested

Line 109: "Bacterial agents were generally more frequently.." How much of this is due to the sample extraction process? Some viruses on the authors' list are RNA viruses, which are more susceptible to degradation.

AU: as indicated earlier, the extraction kit used allows to extract DNA and RNA so we are confident that the extraction process does not strongly impact our results.

Comments on the Quality of English Language A thorough grammar check is recommended.

AU: the manuscript has been sent for a thorough English revision by a native English speaker.

Round 2

Reviewer 2 Report

Comments and Suggestions for Authors

Thank you for considering my comments into account. Nevertheless, I still have some suggestions that I firmly recommend to take into account. The objective is still expressed as “to report results” and this is not a scientific question from a formal point of view. I suggest changing it by some something like this: “The objective of the current study was therefore to report the results investigate the frequency of 10-pathogens by multiplex RT-qPCR on submitted samples for BRD diagnosis from a 5-year period of a 10-pathogen used in to a diagnostic laboratory (Biovet inc.) in the Province of Quebec…

The term prevalence is still being used in the manuscript. See pag 4, line 158, 170; pag. 14, line, 472, 475, 477, and Footnote figure 6 (line 406). Please, revise all the text.

 About the use of Figure 6 in the discussion, thank you for being open to considering my opinion. Without any intention of polemic, I am convinced that it is inappropriate to present this figure as a support in the discussion without having previously presented it in results and discussed it. Comparing the relative frequencies of pathogens associated with BRD in respiratory samples from different studies would be the objective of a systematic review (in which the results could be presented with this type of figure, as it is very visual, and discussed). However, this is not the aim of this study nor what has been done in it. The discussion simply mentions the studies that the authors consider relevant for interpreting their results, which is appropriate. Therefore, I recommend that the figure be removed, and the references cited in the text.

Author Response

REVIEWER 2

Thank you for considering my comments into account. Nevertheless, I still have some suggestions that I firmly recommend to take into account. The objective is still expressed as “to report results” and this is not a scientific question from a formal point of view. I suggest changing it by some something like this: “The objective of the current study was therefore to report the results investigate the frequency of 10-pathogens by multiplex RT-qPCR on submitted samples for BRD diagnosis from a 5-year period of a 10-pathogen used in to a diagnostic laboratory (Biovet inc.) in the Province of Quebec…

Authors (AU): we thank you for your comments. We have highlighted in yellow the changes in the manuscript. We agree with the suggested change which appears now in L24-26.

The term prevalence is still being used in the manuscript. See pag 4, line 158, 170; pag. 14, line, 472, 475, 477, and Footnote figure 6 (line 406). Please, revise all the text.

AU: sorry for this mistake. We have corrected it as suggested throughout the revised version of the manuscript and also changed the x-axis legend of figure 4.

 About the use of Figure 6 in the discussion, thank you for being open to considering my opinion. Without any intention of polemic, I am convinced that it is inappropriate to present this figure as a support in the discussion without having previously presented it in results and discussed it. Comparing the relative frequencies of pathogens associated with BRD in respiratory samples from different studies would be the objective of a systematic review (in which the results could be presented with this type of figure, as it is very visual, and discussed). However, this is not the aim of this study nor what has been done in it. The discussion simply mentions the studies that the authors consider relevant for interpreting their results, which is appropriate. Therefore, I recommend that the figure be removed, and the references cited in the text.

 AU: We agree with the reviewer’s point of view. We have removed the figure from the revised version and have included the references in the discussion (L444-449).

We sincerely hope that this revised version of the manuscript is now acceptable for publication.

Reviewer 3 Report

Comments and Suggestions for Authors

In this revised version, the authors address most of the comments raised in the previous round and make statements that explicitly state the study's limitations. As a result, the current version has improved in quality. A minor comment: The authors should pay attention to using consistent fonts and font sizes throughout the manuscript since such inconsistencies can distract readers; for example, Tables 1 and 2 font sizes are too small. In Figure 3a, please indicate how the ranges in Ct correspond to varying degrees of positivity as per Table 1 -- this can be shown with labels like "Negative," "Weak Positive," etc., on the top. In Figure 3C, please label each color legend for clarity (a suggestion from the last round). Currently, it just reads as "values." 

Author Response

REVIEWER 3

In this revised version, the authors address most of the comments raised in the previous round and make statements that explicitly state the study's limitations. As a result, the current version has improved in quality. A minor comment: The authors should pay attention to using consistent fonts and font sizes throughout the manuscript since such inconsistencies can distract readers; for example, Tables 1 and 2 font sizes are too small. In Figure 3a, please indicate how the ranges in Ct correspond to varying degrees of positivity as per Table 1 -- this can be shown with labels like "Negative," "Weak Positive," etc., on the top. In Figure 3C, please label each color legend for clarity (a suggestion from the last round). Currently, it just reads as "values." 

Authors (AU): we thank you for your comments. We have highlighted in yellow the changes in the manuscript. We have tried to increase the font sizes of the tables 1 and 2 as requested.

We also changed the figure 3 as requested (adding the interpretation based on Ct in 3A and adding color legend in 3C).

We sincerely hope that this revised version of the manuscript is now acceptable for publication.